# HYPERNETWORK-BASED EQUIVARIANT CNNS

## ABSTRACT

In geometric deep learning, numerous works have been dedicated to enhancing neural networks with ability to preserve symmetries, a concept known as equivariance. Convolutional Neural Networks (CNNs) are already equivariant to translations. To further achieve rotation and reflection equivariance, previous methods are primarily based on Group Equivariant Convolutional Neural Networks ($G$-CNN). While showing a significant improvement when processing rotation-augmented datasets, training $G$-CNN on a dataset with little rotational variation typically leads to a performance drop comparing to a regular CNN. In this study, we discuss the reason of $G$-CNN not performing on datasets with little rotational variation. We propose an alternative approach: generating CNN filters that inherently exhibit rotational equivariance without altering the main network's CNN structure. This is achieved through our novel application of a dynamic hypernetwork. We prove these generated filters grant equivariance property to a regular CNN main network. Our experiments demonstrate that our method outperforms $G$-CNN and achieves performance comparable to advanced state-of-the-art $G$-CNN-based methods.

## 1 INTRODUCTION

In machine learning, convolutional neural networks (CNNs) are effective and prevalent tools for image classification and segmentation tasks. When a specific area of the image input is translated to a different location, the convolution operation inherently causes the extracted feature to translate similarly. This ability to comprehend and preserve translation is known as translation equivariance, and general equivariance is highly significant in geometric deep learning. Wood & Shawe-Taylor (1996) emphasize that it is a central problem to design neural networks that exhibit invariance or equivariance to representations in machine learning. It is a natural question to ask: Can a CNN exhibit equivariance regarding rotations and reflections?

**Group Equivariant Convolutional Networks** ($G$-CNNs), introduced by Cohen & Welling (2016), is one of the most popular equivariant CNN structure. The convolution operation is modified to be group equivariant and demonstrated equivariance for rotations, translations, and reflections. The $G$-CNN architectures have been the backbone of many equivariant neural networks, including steerable CNNs (Cohen & Welling, 2017; Weiler et al., 2018) and spherical CNNs (Cohen et al., 2018; Salihu et al., 2024), with several applications such as domain adaptation (Zhang et al., 2022), pose estimation (Howell et al., 2023; Li et al., 2021), and many other areas involving symmetries.

$G$-CNN's equivariance property enables effective handling of data with symmetries, but equivariance comes with an **additional constraints** imposed on the filters: Filters are grouped into smaller subsets, where the filters in each subset are strictly related by reflections and rotations. When training on a dataset with little rotational variation, these constraints can lead to $G$-CNNs underperforming compared to standard CNN architectures. To empirically observe this, in Table 1, we train a small CNN and a small $G$-CNN on the original MNIST datasets. CNN outperforms $G$-CNN on the original MNIST testing dataset, while $G$-CNN outperforms CNN when the testing dataset is incorporated with rotational augmentation.[1]

To bypass such constraint within the filters, in this paper we propose a Hypernetwork-based Equivariant CNN (HE-CNN) to achieve equivariance. The proposed HE-CNN is grounded in a straightforward concept: rotating filters in alignment with input rotation produces corresponding rotated features.

---

[1]Experimental details are presented in Section 5.

Specifically, HE-CNN consists of a dynamic hypernetwork and a main network. The main network can be a general CNN and the dynamic hypernetwork generates input-dependent parameters. The dynamic hypernetwork is composed of two components: a non-equivariant parameter-pieces (NEP) generator and a novel module called equivariant combiner. The NEP generator generates parameter-pieces of entire parameters. The equivariant combiner, abbreviated as equi-combiner, combines the parameter-pieces in a specific pattern to form full parameters with the ability to follow given symmetries on the inputs. Finally, we theoretically and empirically demonstrate that the proposed HE-CNN confer the equivariance property to non-equivariant CNN structures.

Our main contribution can be summarized as follows.

1. We purpose an alternative way to achieve equivariance: instead of constraining the filters, equivariance is achieved through input-dependent parameters.

2. We propose the HE-CNN to achieve the equivariance.

3. Extensive experiments demonstrate the effectiveness of the proposed HE-CNN.

Table 1: Performance of regular CNN and $G$-CNN trained on the original MNIST.

| Test Dataset | CNN | $G$-CNN |
|---|---|---|
| Original MNIST | **96.29** | 85.88 |
| Randomly Rotated MNIST | 34.81 | **50.02** |
| 90° rotated MNIST | 17.76 | **85.88** |
| Mean of the first two | 65.55 | **67.95** |
| #Parameters | 3370 | 12522 |

## 2 RELATED WORK

***Group Equivariance CNNs (G-CNN)*** are introduced by Cohen & Welling (2016) as one of the earliest adaptation of general equivariance into CNN, and has been the backbone for the majority of equivariant architectures. Assume the group consists of translation and 90 degree rotations. The first layer of $G$-CNN would have a filter in the same fashion of a regular CNN, and rotate this filter by $90°, 180°$ and $270°$, resulting in total four rotated versions of a same filter. These four filters output four feature maps for a single input. For all the following layers, to process these four features, $G$-CNN has four different filters as a set of filters, and this set of filters is again rotated by multiples of 90 degrees to perform the $G$-equivariant convolution. $G$-CNNs apply pooling to the final set of features before the linear layer, achieving invariance in the final output.

A previous approach on making non-equivariant model equivariant is the canonicalization method (Kaba et al., 2023; Mondal et al., 2023). The equivariance is achieved through a $G$-CNN based canonicalization network, learning to rotate the input before feeding into the pretrained model. HE-CNN avoids the $G$-CNN structures, and is utilized during the training process.

***Hypernetwork***, initially introduced by Ha et al. (2016), provides an alternative approach to train a neural network. It is widely used in federated-learning (Shamsian et al., 2021), few-shot learning (Sendera et al., 2023; Yin et al., 2022), continual learning (Hemati et al., 2023) and many other areas, due to its versatility and parameter-efficiency. The network designated for training on a specific dataset is known as the *main* or *target* network. The network responsible for generating the parameters of the main network is referred to as the *hypernetwork*. If the hypernetwork generates input-dependent parameters, we call it a *dynamic* hypernetwork. Otherwise we say it is *static*. The combination of a main network and a hypernetwork is referred to as a *full network* in this paper.

For hypernetwork research related to equivariance, Garrido et al. (2023) uses hypernetwork to make representations stay equivariant without converging to invariance. The hypernetwork is utilized for parameter sharing, not for achieving equivariance. To the best of our knowledge, there is no work to use hypernetworks to achieve the equivariance on a regular CNN structure.

## 3 PRELIMINARY

We give a brief definition of groups and representations, and a comprehensive definition can be found in Section A.1. For a set $G$ with a operation $*$, $(G, *)$ or simply $G$ is a *group* if all following properties are satisfied: $*$ is associative, $G$ has an identity element, and every element has an inverse. Given $G$ and a vector space $V$, a *representation* $(V, \rho)$ or simply $\rho$ is a mapping on $G$, where every $\rho(g)$ is a linear map on $V$. Furthermore, $\rho(g * g') = \rho(g)\rho(g')$ for any $g, g' \in G$.

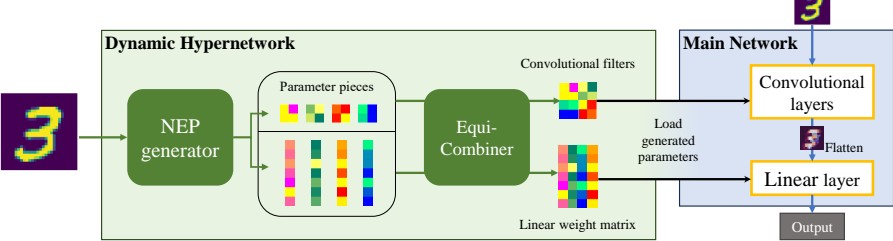

Figure 1: An overview of the HE-CNN architecture, where all the learnable parameters lie in the NEP generator.

If $\rho$ is fixed and we want to study each $g$, $\rho(g)(x)$ can be shorten to simply $gx$. If we want general properties for all $g \in G$, $\rho x$ stands for $\rho(g)x$ for all $g \in G$. A representation $\rho$ is called a *trivial* representation if it $\rho x = x$ for all $x \in V$.

We fix group $G$ based on the desired symmetries for given tasks. Now we can define equivariance and invariance in deep neural networks.

**Definition 1.** Let $f$ denote a deep neural network or one neural network layer. For two representations $\rho_1$ and $\rho_2$, we say $f$ is *(G-)Equivariant* if for any input $x$, we have

$$\rho_2 f(x) = f(\rho_1(x)).$$

If $\rho_2$ is the trivial representation, then we say $f$ is *(G-)Invariant*. That is,

$$f(x) = f(\rho_1(x)).$$

An equivariant neural network typically comprise multiple equivariant layers followed by an invariant layer prior to processing downstream tasks, a structure that has demonstrated benefits in various research contexts (Cohen & Welling, 2016; Worrall et al., 2017; Wang et al., 2022). We also adopt this architectural principle in the design of our models.

When utilizing hypernetwork on a main neural network $f$, the input space is denoted by $X$, and the weight space in neural networks is denoted by $\Omega$. We denote a dynamic hypernetwork by $w : X \to \Omega$. For any input $x \in X$, the corresponding generated weight is $w(x)$, and $f_{w(x)}$ refers to the main network that loads $w(x)$ as its parameters. The full network is denoted by $f_{w(\cdot)}(\cdot)$, or simply $f_w$, which maps input $x$ to $f_{w(x)}(x)$. Within the context of dynamic hypernetworks, $f_{w(\cdot)}(\cdot)$ is $G$-equivariant if $\rho_2 f_{w(x)}(x) = f_{w(\rho_1 x)}(\rho_1 x)$, and invariant if $\rho_2$ is the trivial representation.

## 4 METHODOLOGY

In this section, we present the proposed HE-CNN model.

### 4.1 THE ARCHITECTURE

We assume input images are square-shaped.[2] As illustrated in Figure 1, the proposed HE-CNN consists of a main network $f$ and a dynamic hypernetwork $w$. The main network $f$ of HE-CNN consists of several convolutional layers, a single flatten layer, several linear layers, and some activation/pooling/batch-normalization layers in between. We fix the group $G$ to be $(Z_4, +_{mod\ 4})$, the group of 90-degree rotations. More general group of rotations, reflections and translations are discussed in Section 4.5. We aim to achieve equivariance on the full network $f_w$, combining $f$ with the input-dependent parameters generated by a dynamic hypernetwork $w$. We summarize our objectives in the following definition.

**Definition 2.** We say $f$ is *w-based equivariant* or *hypernetwork-based equivariant* if the following conditions hold:

---

[2]Due to their correspondence with rotations, input images of equivariant structures are typically selected to be square-shaped (Cohen & Welling, 2016).

1. For any fixed parameters $\gamma$, $f_\gamma$ is not equivariant.

2. When using $w$ to generate input-dependent parameters, $f_{w(\cdot)}(\cdot)$ is equivariant to $G$.

In other words, $f$ is $w$-based equivariant if $w$ grants equivariance to non-equivariant main network. The output of $w$ is referred to as equivariant parameters.

The objective of the proposed HE-CNN is to ensure $w$ generates equivariant parameters. This $w$ consists of two components. The first component is a *non-equivariant parameter-pieces generator (NEP generator)*. NEP generator is responsible for generating approximately $1/4$ of the total parameters, including filters in convolutional layers and weight matrices for linear layers. The details for each case are introduced in the next two sections. The NEP generator can be any non-equivariant neural network and our implementation uses regular CNNs. Additionally, we expect it to be non-equivariant to avoid collapsing to invariant filters as demonstrated in Section D.1.

When given an input image, we collect the four $90°$-rotated versions of it and send them to the NEP generator, resulting in four *parameter pieces*. Then, the four parameter pieces are fed into the second component, the *equivariant combiner (equi-combiner)*, to assemble parameter pieces to get full parameters that could achieve the equivariance.

The design of the equi-loader is outlined in the following two sections, which correspond to two cases: convolutional layers and linear layers. As explained in Section 3, we follow previous designs (Cohen & Welling, 2016; Worrall et al., 2017; Wang et al., 2022) that for the convolutional layers, the objective of the equi-combiner is to ensure the equivariance and preserve the rotations. For linear layers, the equi-combiner is to generate invariant parameters, so that the final output remains consistent regardless of the input's rotations. Hence, in the next two sections, we discuss how to design the NEP generator and equi-combiner for the two types of layers.

## 4.2 Hypernetwork for Convolutional Layers

For simplicity, we assume that there is only one convolutional layer in the main network $f$ and if there are multiple ones, we can apply the same operation to each of them. In the following, we discuss how to design the hypernetwork for generating parameters in this convolutional layer.

The group $Z_4$ has four elements: 0,1,2 and 3. They corresponds to $0°$, $90°$, $180°$ and $270°$ accordingly. The chosen permutation $\rho(g)$ is to perform counter-clockwise rotations corresponding to $g$ on input image $x$. For all the convolutional layers, we let both $\rho_1$ and $\rho_2$ in the definition of equivariance to be the same $\rho$. For simplicity, we write $\rho(g)x$ as $gx$. The NEP generator is denoted by $N$.

For the convolutional layer in $f$ has input channels $C_{\text{in}}$, output channels $C_{\text{out}}$ and filter dimension $K$, the weight of the convolution filter is in shape $(C_{\text{out}}, C_{\text{in}}, K, K)$ and the bias is in shape $(j)$. The output shape of $N$ is $(C_{\text{out}}, C_{\text{in}}, \lceil K/2 \rceil, \lceil K/2 \rceil) + (C_{\text{out}})$, where the first part is for parameter pieces of filters, the second part is for the bias, and $\lceil \cdot \rceil$ denotes the ceiling function. Accordingly, the NEP generator can be split into $N_{\text{filters}}(x)$ for parameter pieces of filters and $N_{\text{bias}}(x)$ for biases. We first study assembling full filters using $N_{\text{filters}}$.

We denote the input by $x^0$, and denote its $90°, 180°$ and $270°$ rotations as $x^1, x^2$ and $x^3$ accordingly. Each of them is sent into $N_{\text{filters}}$. The corresponding outputs (i.e., parameter pieces) are denoted by $a^i = N_{\text{filters}}(x^i)$ $(i = 0, 1, 2, 3)$. The equi-combiner takes each $a^i$, and perform a **clockwise** $i \cdot 90°$ rotation for each and get $a_r^i$. As illustrated in Figure 2, we assemble them in a **clockwise** manner(i.e., we assign $a_r^0$ on the top left, $a_r^1$ on the top right, $a_r^2$ on the bottom right, and $a_r^3$ on the bottom left) to be the final filter. If the dimension $K$ is odd, we average the $(K+1)/2$-th and $(K-1)/2$-th columns of the final filter into one column and perform similar operations on the $(K+1)/2$-th and $(K-1)/2$-th rows. Mathematically, the generation of weights in a filter via the NEP generator can be formulated as

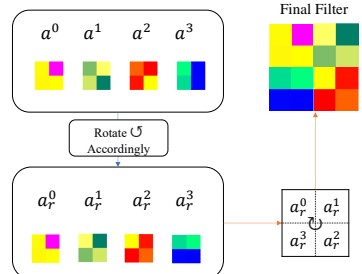

Figure 2: A visualization of the equi-combiner for a 4x4 filter.

$$w_{\text{filters}}(x) = \sum_{g \in Z_4} g^{-1} E\big(N_{\text{filters}}(gx)\big),$$

where $E$ is the expansion operation, placing parameter pieces to the top left corner of a filter, and divide the middle columns and rows by two if $d$ is odd.

For the bias, we expect it to be identical for all the rotated versions of the input. To achieve that, we simply average the outputs of $\{N_{\text{bias}}(x^i)\}_{i=0}^3$ as

$$w_{\text{bias}}(x) = \frac{1}{4} \sum_{g \in Z_4} N_{\text{bias}}(gx). \tag{1}$$

For the hypernetwork designed above for convolutional layers, we show that it achieve the equivariance on the full network in the following theorem.[3]

**Theorem 1.** *Let $f^{conv}$ be a main neural network with several convolutional layers, and let $w$ be the hypernetwork composed of the NEP generator and the equi-combiner as designed above. Then, for all $g \in Z_4$, we have*

$$g f_{w(x)}^{conv}(x) = f_{w(gx)}^{conv}(gx).$$

For an input $x$, after all convolutional layers of the full network $f_w$, we have an extracted features $\{B_i\}_{i=1}^{C_{\text{out}}} = f_{w(x)}^{conv}(x)$. For simplicity, we first look at the case when $C_{\text{out}} = 1$, and $B = f_{w(x)}^{conv}(x)$. For the next step in HE-CNN, $B$ are flattened and sent into following linear layers of the full network. We denote the flatten operation as $P$, and the flattened vector $v = P(B)$. If the dimension of $B$ is $d \times d$, the length of $v$ is $d^2$. For the general case where $C_{\text{out}} > 1$, we repeat the process for all $B_i$.

### 4.3 HYPERNETWORK FOR LINEAR LAYERS

In this section, we first study the case when the main network $f$ consists of only one linear layer. The linear layer takes flattened vector $v$ as the input, and output a vector of length $m$. In this case, the weight matrix $W$ of the main network $f$ is in shape $(m, d^2)$. The hypernetwork $w$ aims to generate input-dependent $W$ so that the output of $f_w$ is invariant, i.e., $f_w(P(B)) = f_w(P(\rho B))$.

Prior to designing such $w$, we need to establish two essential propositions. First, we adapt the rotation representation $\rho$ for the square matrix from Section 4.2 to the flattened vectors $v$ using $\rho_{\text{lin}} = P\rho P^{-1}$. Now, we can write the above condition for invariance on $f_w$ as $f_w(v) = f_w(\rho_{\text{lin}}v)$.

The following proposition allow us to view $\rho_{\text{lin}}$ as a collection of length-4 permutations.

**Proposition 2.** *For the representation $\rho_{lin}$ on the vector $v$, we have the following:*

1. *Let $v' = \rho_{lin}(1)v$ be $\rho_{lin}$ applied on $v$ once. For any entry $v_i$ of the vector $v$, there is an unique permutation $\sigma = (i, j, k, l)$ such that $v_i = v'_j$, $v_j = v'_k$, $v_k = v'_l$, and $v_l = v'_i$.* [4]

2. *The representation $\rho_{lin}$ can be viewed as a collection of all such length-4 permutations on the index of $v$.*

Since $v$ is length-$d^2$, we have $\lceil d^2/4 \rceil$ amount of different length-4 permutations.[5] A single permutation on $v$ is illustrated on top of Figure 3. We denote the collection of all such permutations as $S$. For simplicity, we fix a $\sigma = (i, j, k, l)$. After the permutation on the indices of $v$, we write $v' = \sigma(v)$. This permutation $\sigma$ can also permute

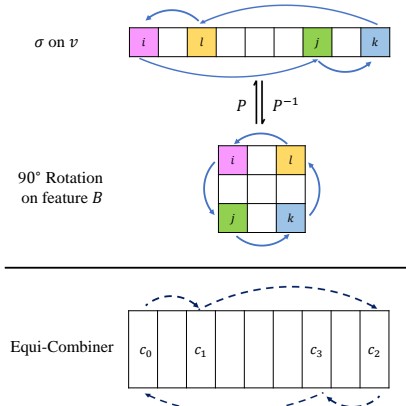

Figure 3: On the top, we visualize a permutation $\sigma = (i, j, k, l)$, and the corresponding 90° rotation. On the bottom, the equi-combiner assign column vectors $\{c_i\}_{i=0}^3$ **reverse** to $\sigma$.

---

[3]All the proofs are presented in Section D.3.

[4]A permutation $(i, j, k, l)$ is a circular expression, that is, $(i, j, k, l) = (j, k, l, i) = (k, l, i, j) = (l, i, j, k)$. Given $i$, the exact expression of $j$, $k$ and $l$ are given in Section D.2.

[5]When $d$ is odd, we have $(d^2 - 1)/4$ amount of length-4 permutations, and the middle point $v_z$ of $v$ is unchanged. By a slight abuse of notation, we can also view the unchanged point as a permutation $(z, z, z, z)$.

the column vectors of the weight matrix $W$, and we denote the new matrix after permutation as $W' = \sigma(W)$.

In the case of convolutional filters, processing rotated inputs with rotated filters results in rotated features. The next proposition shows that we have a similar result for linear weights.

**Proposition 3.** *In the main network $f$, denote the input vector of the linear layer as $v$, and denote the weight matrix of the linear layer as $W$. For any given permutation $\sigma$, if $v' = \sigma(v)$ and $W' = \sigma(W)$, we have*

$$v'W'^{\top} = vW^{\top}, \tag{2}$$

*where the superscript $^{\top}$ stands for the transpose operation.*

Eq. (2) is exactly the affine linear operation in the linear layer of $f$ without the bias. Therefore, we intend for the hypernetwork $w$ to generate weight matrix $W$ that permutes its column vectors as $v$ is permuted by $\sigma$. If the linear layer in the main network $f$ requires bias, $w$ should generate same bias regardless of the permutation.

To accomplish this, our NEP generator $N$ generate one column vector per permutation $\sigma \in S$ and all of the bias, resulting in the output in shape $(m, \lceil d^2/4 \rceil) + (d^2)$. We can again split the output into $N_{\text{weight}}(x)$ and $N_{\text{bias}}(x)$. Furthermore, for each $\sigma \in S$, denote the corresponding generated column vector in $N_{\text{weight}}(x)$ by $N_\sigma(x)$. We fix a $\sigma = (i, j, k, l) \in S$ to illustrate the equi-combiner. If there are more than one $\sigma \in S$, we repeat the process for all of such $\sigma$.

Given an input image $x^0$, we denote all rotated versions as $x^1$, $x^2$, and $x^3$. All four $x^i$ are fed into the NEP generator $N_\sigma$, and denote the output column vectors by $c_i = N_\sigma(x^i) \in \mathbb{R}^{m \times 1}$ for $i = 0, 1, 2, 3$. The equi-combiner assign $\{c^i\}_{i=0}^3$ into the parameter matrix $W$ in the direction of $\sigma^{-1}$. That is,

$$W_i = c_0, W_l = c_1, W_k = c_2, W_j = c_3, \tag{3}$$

where $W_n$ denotes the $n$-th column of $W$. On the bottom of Figure 3, we give an illustration of the equi-combiner for a single $\sigma$. We can repeat the above assembling process for all permutations in $S$ to obtain full $W$. [6]

Given four column vectors $\{c_i\}_{i=0}^3$ and the corresponding length-4 permutation $\sigma$, Eq. (3) above defines a operation $E_{\sigma^{-1}}$ which expand column vectors $\{c_i\}$ into the weight matrix $W$. Hence, the generation in the dynamic hypernetwork can be expressed as

$$w_{\text{weight}}(x) = \sum_{\sigma \in S} E_{\sigma^{-1}}\big(\{N_\sigma(gx)\}_{g \in Z_4}\big).$$

Similar to the convolutional case, we expect the bias to be the same for all inputs and so we collect the output of the NEP generator and simply take the average as in Eq. (1).

For the above design, we prove the invariance property for the linear layer in the full network $f_w$ in the following theorem.

**Theorem 4.** *Let $f^{lin}$ be one single linear layer. Let $w$ be the hypernetwork defined above. Then, $w$ generate invariant parameters for $f^{lin}$. That is, for all input $x$ and $g \in Z_4$, we have*

$$f_{w(x)}^{lin}(x) = f_{w(gx)}^{lin}(gx).$$

We have shown invariant outputs after the first linear layer. For the rest of the linear layers (if any), all weights should be the same regardless of the permutation to preserve the invariance of the full network $f_w$. To do that, for any additional layers, NEP generate outputs full weight $W$ for any of the rotated inputs $\{x^i\}$, and we simply average all outputs to get the weight matrix.

## 4.4 PROPERTIES OF HE-CNN

Based on the above designs of HE-CNN, we can prove that the proposed HE-CNN can achieve equivariance for intermediate features, and invariance for the final output.

---

[6]When $d$ is odd, the middle point $v_z$ of $v$ is unchanged. Based on Eq. (3), all generated column vectors $c_i$ are assigned into $W_z$. For consistency, all four column vectors are averaged into one.

**Theorem 5.** *Let $f$ be a main neural network with several convolutional layers and several linear layers. Let $w$ be a hypernetwork for both convolution layers and linear layers. Then, $f$ is $w$-based invariant. That is, for any $g \in Z_4$,*

$$f_{w(x)}(x) = f_{w(gx)}(gx).$$

Moreover, for other commonly used layers (e.g., pooling, activation and batch-normalization layers) in CNNs, our equivariance and invariance is unaffected.

**Theorem 6.** *Let $f$ be a main neural network with several convolutional layers, and linear layers. We have achieved $w$-based equivariance on $f_w$. If we insert pooling layers, activation layers, and batch-normalization layers in $f_w$, our equivariance still holds.*

### 4.5 EXTENSION TO GROUP $D_{4n} \times \mathbb{R}^2$

We have achieved $Z_4$-equivariance, and CNN structure is already equivariant to $\mathbb{R}^2$. Now we extends to $D_{4n}$, the group of reflections and $^{90}/n^\circ$ rotations.

As a proposition in Basu et al. (2023), given an arbitrary group $G$ and a non-$G$-equivariant neural network $N$, $F(x) = \sum_{g \in G} \rho_g^{-1} N(\rho_g(x))$ is always equivariant regarding $G$.

We choose $G$ to be the quotient group $D_{4n}/Z_4$. Since our $f_{w(\cdot)}$ is already equivariant to $Z_4$ and translation, it is trivial to see that the following expression is equivariant to $D_{4n} \times \mathbb{R}^2$:

$$F_{w(x)}(x) = \frac{1}{n} \sum_{g \in D_{4n}/Z_4} g^{-1} f_{w(gx)}(gx).$$

After extending to $D_{4n} \times \mathbb{R}^2$, $F_{w(\cdot)}(\cdot)$ is named $D_{4n}$-HE-CNN.

### 4.6 TRAINING PROCESS

Given a fixed main network, we first initialize the NEP generator $N$. We choose a CNN structure with output dimension described in Sections 4.2 and 4.3. During training, the original inputs are sent into the combination of NEP generator and equivariant combiner to generate equivariant parameters. The main network $f$ then loads the equivariant parameters and process the same input. We compute the chosen loss between the output of $f_w$ and the label, and preform back propagation to update the learnable parameters in the NEP generator.

When the inputs are in batch, our hypernetwork is indeed capable of generating parameters in batch. However, for inputs in batch size $b$, we would have $b$ corresponding parameters, thus $b$ main networks. This bijection can easily lead to GPU memories shortage when main network is large. We comprehend this by the following: for each batch of inputs, we average them to get one single set of parameters, used to process the batch of inputs. This significantly reduced the memory usage. However, due to the average operation, our network is equivariant if the representation $\rho(g)$ is applied on the whole batch. For each experiment, we specify whether we use parameters in batch or averaged.

### 4.7 UTILIZING MULTI-HEAD AND LoRA

For the NEP generator $N$, the output size of the last layer is approximately one forth of the parameter of the main network. Assuming the main network $f$ has $l$ parameters, the parameters in the last linear layer of $N$ are $O(l)$. This can become excessive if the main network is significantly large.

To reduce the parameter count for $N$, we utilize the low-rank adaptation (Hu et al., 2021) on the last linear layer. We choose an intermediate rank $r$. On the last linear layer, instead of directly generating the full length $l$, we generate two outputs, each in shape $\sqrt{l} \times r$ and $r \times \sqrt{l}$. Multiplying them would give us our needed length $l$ parameters. Now, the parameters of $N$ with LoRA is $O(\sqrt{l})$.

During our experiments, we discovered that applying LoRA directly can lead to a significant and undesirable drop in the performance of HE-CNN. Therefore, for each layer of the main network, we have a corresponding linear head in the parameter generator to generate layer-specific parameter. Then, we apply LoRA on each linear head. Each head has different $l$ depending on the structure

of the main network, but they share the same rank $r$. Notice that we only divide the linear layer of $N$ into smaller linear heads. The shared convolutional layers remain unchanged compared to the scenario without LoRA.

## 5 EXPERIMENT

We first use a simple network to empirically show the affect of $G$-CNN filters constraint. Then, we empirically evaluate our model using several benchmark datasets and compare its performance against base $G$-CNN structures and state-of-the-art equivariant methods.

### 5.1 IMPACT OF G-CNN'S CONSTRAINTS ON FILTERS

Table 2: Classification accuracy (%) of small models trained and tested on different versions of MNIST. A superscript of $o$ or $r$ implies the model is trained on the original or random rotated MNIST.

| Test Dataset | $CNN^o$ | $Z_4$-$CNN^o$ | $Ours^o$ | $CNN^r$ | $Z_4$-$CNN^r$ | $Ours^r$ |
|---|---|---|---|---|---|---|
| Original MNIST | **96.29** | 85.88 | 96.03 | 75.27 | 76.22 | **96.44** |
| Randomly Rotated MNIST | 34.81 | 50.02 | **76.94** | 73.08 | 75.51 | **96.32** |
| 90° rotated MNIST | 17.76 | 85.88 | **96.03** | 75.68 | 76.22 | **96.44** |
| Mean of first two | 65.55 | 67.95 | **86.49** | 74.18 | 75.87 | **96.63** |
| Equivariance difference | 68.86% | 0% | 0% | 0.27% | 0% | 0% |
| # Parameter | 3370 | 12522 | 11527 | 3370 | 12522 | 11527 |

As discussed in Section 1, $G$-CNN structure requires additional constraints on filters to achieve equivariance. The first experiment is to demonstrate the affect of such constraints, and compare HE-CNN with CNN and $Z_4$-CNN. The network is intentionally chosen to be simple for effective demonstration, and the group $G$ is chosen to be $Z_4$. Both CNN model and $Z_4$-CNN model have 3 hidden layers with 16 hidden channels. For HE-CNN, we use the same CNN structure as the main network $f$, and the NEP generator $N$ used multi-head and LoRA. To keep the parameter count close to $Z_4$-CNN, $N$ is chosen to have 2 hidden convolutional layers with 16 hidden channels. All datasets are divided into non-overlapping training and testing subsets with a 6:1 ratio. The Rotated MNIST dataset is derived from the original MNIST dataset by applying random rotation augmentation to each image. We also manually rotate all images in the original MNIST by exactly 90° to get 90° rotated MNIST for testing model's equivariance property.

Each model is either trained on the original MNIST or randomly rotated MNIST, indicated by superscripts of $o$ or $r$ accordingly. Each trained model is tested on the original, randomly rotated, and 90°-rotated MNIST, and the performance is presented in Table 2. We further displayed the average accuracy on both the original and randomly rotated datasets to demonstrate consistent performance across regular and rotated data. Additionally, we compute the relative difference (difference divided by sum) of the accuracy on the original and 90°-rotated MNIST as *equivariance difference*. A lower percentage indicates a better understanding of 90° rotations, with 0% representing strict equivariance. The experiment result illustrates that, in terms of average performance and performance on rotated dataset, $Z_4$-CNN outperforms CNN in both training scenarios: the original MNIST and the randomly rotated MNIST. However, due to the constraint on $Z_4$-filters, when trained and tested on original MNIST, $CNN^o$ outperforms $Z_4$-$CNN^o$. The HE-CNN shows better performance in almost all settings, with the only exception when trained and tested on original MNIST, where we have a comparable result with $CNN^o$. This empirically shows we bypass the constraint while achieving equivariance.

### 5.2 CNN ON ROTATED-MNIST

For classification on the Rotated MNIST dataset, the main network is composed of seven convolutional layers and two linear layers, similar to the CNN used by Cohen & Welling (2016). Since the main network is rather small, we use generated parameters in batch to process batched input. The NEP generator $N$ is composed of three convolutional layers and one linear layer. When using LoRA and multi-head, we choose the intermediate rank $r$ to be 4. Since equivariant structures benefit from rotation augmentation that is not included in the chosen group $G$, we train HE-CNN on the rotated MNIST training dataset.

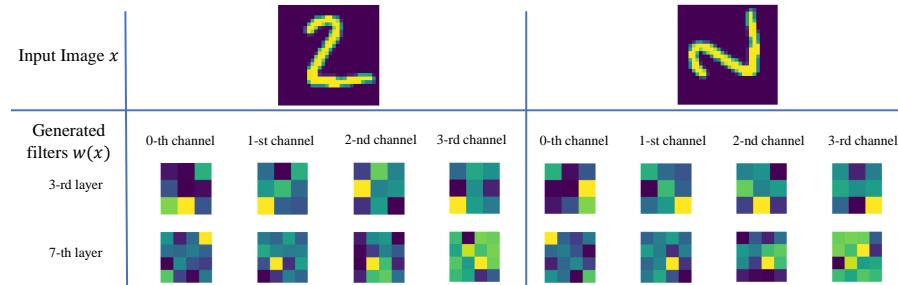

Figure 4: For two images related through 90° rotations, we visualize generated filters through the hypernetwork $w$.

Results are shown in Table 3. The full model outperforms previous state-of-the-art method, while HE-CNN with LoRA ensure a better performance comparing to the original $G$-CNN method.

Furthermore, we provide a visualization of generated filters for an MNIST image $x$ and its rotated version $x'$ in Figure 4. Numerically, we rotate the generated filters of $x'$ back, and compare with filters generated by $x$. The average MSE difference between two sets of filters is less than $10^{-7}$, indicating equivariance under floating-point computations.

Table 3: Classification Accuracy on the Rotated-MNIST.

| Model | Accuracy (%) |
|---|---|
| **Regular CNN based Methods** | |
| Sohn & Lee (2012) | 95.8 |
| Schmidt & Roth (2014) | 96.02 |
| **Equivariance based Methods** | |
| Z2-CNN (Cohen & Welling, 2016) | 94.97 |
| P4-CNN (Cohen & Welling, 2016) | 97.72 |
| LieConv (Finzi et al., 2020) | 98.76 |
| Steerable-CNN (Weiler et al., 2018) | 99.27 |
| E2FCNN (Weiler & Cesa, 2019) | 99.32 |
| Sim2-CNN (Knigge et al., 2022) | 99.41 |
| **Hypernetwork based Methods** | |
| Z4-HE-CNN | **99.50** |
| Z4-HE-CNN(LoRA) | 97.91 |
| D8-HE-CNN(LoRA) | 98.01 |
| D16-HE-CNN(LoRA) | 98.05 |

## 5.3 RESIDUAL NETWORKS AND DEEPER CNNS ON CIFAR10/100

For CIFAR10 and CIFAR100 datasets, we compare HE-CNN's performance with $G$-CNN. Additionally, partial equivariance (Romero & Lohit, 2022) is a follow-up work based on $G$-CNN. Based on the data, partial equivariance $G$-CNN learns additional information of whether certain symmetries are beneficial or harmful. By only keeping the beneficial symmetries, partial equivariance $G$-CNN loose the constraint on the filters. However, it is not strictly equivariant, rather equivariant to a subset of the angles. In comparison, HE-CNN remains strictly equivariant to the chosen group, and need no additional learning on beneficial symmetries.

Table 4: Classification Accuracy (%) for CIFAR10 and CIFAR100 on a residual network.

| Symmetry Group | Model | CIFAR10 | CIFAR100 |
|---|---|---|---|
| $\mathbb{R}^2$ | Residual network | 83.11 | 47.99 |
| $Z4 \times \mathbb{R}^2$ | $G$-CNN | 83.73 | 52.35 |
| | Partial equivariance | **86.15** | **53.91** |
| | HE-CNN | 85.99 | 53.56 |
| | HE-CNN(LoRA) | 83.92 | 52.80 |
| $D8 \times \mathbb{R}^2$ | $G$-CNN | 85.55 | 55.55 |
| | Partial Equivariance | **89.00** | **57.26** |
| | HE-CNN | 88.67 | 56.95 |
| | HE-CNN(LoRA) | 86.34 | 55.73 |

Table 5: Classification Accuracy (%) for CIFAR10/100 on 13-Layer CNN (Laine & Aila, 2017).

| Symmetry Group | Model | CIFAR10 | CIFAR100 |
|---|---|---|---|
| $\mathbb{R}^2$ | 13-Layer CNN | 91.21 | 67.14 |
| $Z4 \times \mathbb{R}^2$ | $G$-CNN | 89.73 | 65.97 |
| | Partial Equivariance | **92.28** | **69.83** |
| | HE-CNN | 91.95 | 66.38 |
| | HE-CNN(LoRA) | 90.12 | 66.14 |
| $D8 \times \mathbb{R}^2$ | $G$-CNN | 90.55 | 67.70 |
| | Partial Equivariance | 91.99 | **70.80** |
| | HE-CNN | **92.07** | 68.89 |
| | HE-CNN(LoRA) | 90.64 | 68.11 |

We test HE-CNN on two different main network structures. The first main network is a residual network, composed with 2 residual blocks of 32 channels each. The second network is a deeper 13-layer CNN used by Laine & Aila (2017). For $G$-CNN and partial-equivariant-$G$-CNN (Partial equivariance), we use the accuracy reported in Romero & Lohit (2022). For HE-CNN, we use hypernetwork $w$ to generate equivariant parameters for convolutional layers and linear layers in both main networks. The NEP generator $N$ is a 3-layer CNN. Due to the size of the main networks, we take average to generate one main network per input batch. When LoRA and multi-head is used, the intermediate rank is set to 7. As demonstrated in Table 4 and 5, we showed comparable results with partial equivariance, while exceeding performance of base $G$-CNN in all settings.

## 5.4 WIDE RESIDUAL NETWORKS ON STL10

We test HE-CNN's ability of handling larger images, using the STL10 dataset with $96 \times 96$ pixels in size. We use the labeled part of the dataset and split into 80-20 training testing ratio. We choose the base model as the wide residual network WRN16/8 by Zagoruyko & Komodakis (2016). For comparison, we replace all convolutional layer with G-equivariant convolutions to get $Z_4$-WRN16/8 and $Z_8$-WRN16/8. On STL10 classification, current state-of-art equivariant model is the $E(2)$-equivariant Steerable CNN (Weiler & Cesa, 2019). It based on steerable CNN (Cohen & Welling, 2017), which extends the concept of $G$-CNN to continuous groups. Their

Table 6: Classification accuracy on STL10, based on Wide-ResNet WRN16/8.

| Model | Accuracy (%) |
|---|---|
| Base WRN16/8 | 87.26 |
| **G-Equivariant Convolutions** | |
| $Z_4$-WRN16/8 | 87.89 |
| $Z_8$-WRN16/8 | 88.87 |
| ***E(2)*-Steerable Convolutions** | |
| WRN16/8-$\{D_8 D_4 D_1\}$ | **90.20** |
| **Hypernetwork-based** | |
| $Z_4$-HE-WRN16/8 | 90.08 |
| $Z_8$-HE-WRN16/8 (LoRA) | 89.13 |
| $D_{16}$-HE-WRN16/8 (LoRA) | 89.75 |

model is denoted by WRN16/8-$\{D_8 D_4 D_1\}$, where each $D_n$ is the group for steerable filters.

For HE-CNN, hypernetwork generates parameters of all convolutional layers and linear layers. The NEP generator consists four consists of four convolutional layer with relu, batch normalization and pooling layers. Due to the size of WRN16/8, we take average and generate one network per batch. When LoRA is used, the intermediate rank is set to 10.

Similar to the case of CIFAR10, our model outperforms regular $G$-CNN, and showed a comparable performance with steerable-CNN based state-of-the-art.

## 6 CONCLUSIONS

In this study, we propose a novel hypernetwork-based equivariant CNN, offering an alternative approach to equivariance without imposing constraints on the filters. Equivariance is achieved through a dynamic hypernetwork composed of a non-equivariant parameter-pieces generator and equivariant combiner. We test our model on several benchmark datasets. Comparing to $G$-CNN based state-of-the-art methods, our network showed either better or comparable results. We showed better performance in all settings compared to the base $G$-CNN. In our future work, we are interested in extending HE-CNN to more complex groups.

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

APPENDIX

# A   ADDITIONAL DEFINITIONS

## A.1   GROUPS AND REPRESENTATIONS

**Definition 3.** Let $G$ be a set and $*$ be an operation. Then, $(G, *)$ is a *group* if the following holds:

1. There exists a $e \in G$, such that for any $g \in G$, $g * e = e * g = g$. We call this $e$ the identity.

2. For any element $g \in G$, there exists $h \in G$ such that $g * h = h * g = e$. We call this the inverse element, and denote it as $-h$ or $h^{-1}$ depending on the context of the operation.

3. $G$ is closed under this operation. That is, for any $g_1, g_2 \in G$, $g_1 * g_2$ is always in $G$.

4. For any $g, h, k \in G$, $g * (h * k) = (g * h) * k$.

After defining a group $(G, *)$, it is common to simply refer to it as $G$ when the operation is clear from the context.

**Definition 4.** Given a group $(G, *)$, a *representation* of $G$ on a vector space $V$ is a map $\rho$ with inputs in $G$. For any $g \in G$, $\rho(g)$ is a *linear map* on $V$. Furthermore, $\rho(g_1 g_2) = \rho(g_1)\rho(g_2)$. We denote the representation by $(\rho, V)$ or simply $\rho$.

The definition of a group and representation might be a bit hard to understand without some background on abstract algebra. It is helpful to think of groups as a collection of symmetries, and representation as an *action* of such symmetries on a vector space. Let us go through one example, the cyclic group of order 4, $(Z_4, +_{mod\ 4})$. Readers with experience in group theory can skip the following example section.

## A.2   EXAMPLE CYCLIC GROUP: $Z_4$

The group $Z_4$ has four elements: $\{0, 1, 2, 3\}$, combined the operation of addition modulo four. Comparing the modulo addition in $Z_4$ with counter-clockwise rotations by multiples of 90 degrees, one can see some *similarity* between them. For instance, $2 + 3 = 1 \mod 4$, and a vector ends up in the same place after rotating 180 degrees and then 270 degrees, as it does after a 90-degree rotation. This *similarity* is captured by a representation. Let $V = \{(x, y) \mid x, y \in \mathbb{R}\}$, all 2D vectors. Our chosen $\rho$ maps $g \in Z_4$ to a linear map on $V$ which perform the corresponding rotation around the origin. For instance, pick $2 \in Z_4$. For any $(x, y) \in V$, $\rho(2)$ is a linear map that rotate $(x, y)$ by 180 degrees, i.e. $\rho(2)(x, y) = (-x, -y)$. Formally, $\rho(g)v = \begin{pmatrix} \cos(g\pi/2) & -\sin(g\pi/2) \\ \sin(g\pi/2) & \cos(g\pi/2) \end{pmatrix} \begin{pmatrix} x \\ y \end{pmatrix}$ and one can check that this indeed satisfies the definition of a representation.

# B   FORCING EQUIVARIANCE THROUGH NUMERICAL METHODS IS NOT VIABLE

Before deploying the equi-combiner, we tried to encourage equivariance by adding another rotation loss: During training, we rotate inputs and compute their generated filters. We compute the MSE Loss between such filters and try to minimize it. Denote the hypernetwork by $w$, then we can write the rotational loss as:

$$L_{\text{rot}} = \frac{1}{4} \sum_{g \in Z_4} L_{\text{mse}}\big(w(gx), w((g+1)x)\big).$$

The performance is poor on 90 degree rotations even though the rotation loss dropped significantly. The test accuracy never surpass $50\%$ on MNIST. We hypothesize that numerical methods can only achieve approximate equivariance in filters. However, this numerical approximation may be insufficient due to the inherent sensitivity of the filters. This is the reason we need strict equivariance guaranteed by our equi-combiner.

## C    EXPERIMENTS DETAILS

When expanding $Z_4$-HE-CNN to a larger groups $D_{4n}$, we only have to consider the extra angles in the first quadrant on the unit circle $\{90/n * i\}_{i=0}^{n-1}$. During training, we first train $f_w$ to be $Z_4$ equivariant for the first half of the training process. Then, for input $x$, we collect all input version of $x$ by angles in $\{90/n * i\}_{i=0}^{n-1}$, and sum their outputs by $f_w$ and take average as our final output.

During all experiments, we use the Adam optimizer. We noticed that it is common to observe minimal change in training loss (especially with LoRA) during the first 50-100 epochs, with test accuracy typically beginning to rise after 150-250 epochs. We believe this arises from the complexities associated with learning to generate parameters. Due to the sensitivity of the generated parameters, we are very cautious about increasing the learning rate. If Adaptive average pooling layer is present in the main network and the output shape is set to 1, we modify to 2 to demonstrate our parameter generation for linear layers.

Next, we provide comprehensive details for our experiments. For the NEP generator of all cases, we provide the detail of the convolutional layers of the generator. After all convolutional layers, features are flatten and sent to a linear layer or several linear heads depending on whether LoRA is used.

For the first experiment where we compare CNN, $Z_4$-CNN and HE-CNN, we choose a fairly small model. Both CNN model and $Z_4$-CNN model have 3 hidden layers with 16 hidden channels and 2 by 2 kernels. We use the same CNN as our main network $f$. The NEP generator is chosen to 2 hidden convolutional layers with 16 hidden channels, and utilized multi-head and LoRA. We use the Adam optimizer and choose the learning rate to be 0.001 for CNN and $Z_4$-CNN, and 0.0002 for HE-CNN. Batch size is set to 32.

For the second experiment on MNIST, the main network $f$ is a CNN described in Cohen & Welling (2016). It has seven convolutional layers, six of them has 20 channels with 3x3 filters, and the last convolutional has 20 channels with 4 by 4 filter. Afterwards, $f$ has two linear layers with 100 intermediate channels. For the NEP generator $N$ in the hypernetwork $w$, we choose a 3-layer convolution with [16,32,32] intermediate channels, [3,3,4] filter size and [2,2,1] stride. We set the learning rate as 0.000075 and the batch size is set to 32. If LoRA is utilized, our intermediate rank is set to 4.

For the experiment on CIFAR10/100, the residual main network is composed with 2 residual blocks of 32 channels, each with filter sizes 3, with additional pooling and batch normalizing layers. The main 13-layer CNN is exactly the same as in Laine & Aila (2017). In both scenarios, learning rates are set to 0.000025. Since we average batch of inputs to get one set of parameters, we choose a smaller batch size as 64 to lower the negative impact. When LoRA is used, the intermediate dimension is set to be 7. The NEP generator $N$ is composed of 3-layer convolution with [32,64,64] intermediate channels, with all filter sizes as 3 and stride as 2 for the last layer. Relu, Batch normalization and Max pooling layers with kernel size as 2 and stride as 2 are inserted between convolutional layers.

For the experiment on STL10, the main network is the wide residual network architecture. The hypernetwork generates the convolutional and linear layer, keeping the others unchanged. The learning rate is set to 0.000015 and batch size is 64. If LoRA is used, the intermediate rank $r$ is chosen to be 10. The NEP generator $N$ is composed of 4-layer convolution with [64,128,128,64] intermediate channels, with all filter sizes as 3 and stride as 2 for the last layer. Relu and Batch normalization are inserted between convolutional layers. Additionally, we perform pooling after first three convolutional layers, with kernel size as 2 and stride as 2.

## D    THEORY DETAILS AND PROOFS

### D.1    REQUIRING NON-EQUIVARIANCE IN OUR PARAMETER GENERATOR

The reason we want non-equivariance is we do not want relations among generated $a_r^i$. If the parameter generator is chosen to be equivariant, filters are rotational invariant as demonstrated on the right of Figure 5, which significantly reduce the expressive power.

| Input: | Non-Equivariant Partial parameter generator | | Equivariant Partial parameter generator | |
|---|---|---|---|---|
| Generated filters | 0-th channel | 1-st channel | 0-th channel | 1-st channel |
| 0-th layer | | | | |
| 3-rd layer | | | | |
| 7-th layer | | | | |

Figure 5: Visualizing generated filters using different parameter generator. Filters on the left are ideal. On the right it collapsed to invariance, i.e., all filters are equal to their rotated versions.

### D.2 PERMUTATION DETAILS OF FLATTENED VECTORS

$Z_4$ is cyclic (i.e. generated by one element), so we only have to show our claim holds for $\rho_{\text{lin}}(1)$. For $v_i$ as $i$-th component in $v$, denote $c = \lfloor i/d \rfloor$ and $r$ as the remainder. These two corresponds to the column and row pre-flatten accordingly. Let $j = d(d-1) - dc + r$. Then, $v'_j = v_i$. We repeat this computation on $j$ to get $k$ and $l$. From the properties of permutations we know that, $(i, j, k, l) = (j, k, l, i)$. This means the starting at $i$ or $j$ ends up with the same permutation. This is why we have $d^2/4$ different permutations.

### D.3 PROOF OF OUR MAIN RESULT

Instead of proving Proposition 5, we prove a more general proposition.

**Proposition 7.** *Let $\{a_i\}_{\{0,1,2,3\}}$ be the generated parameter-pieces mentioned in Section 4.2.*

*If the generated filter rotates with the inputs, the designed equi-combiner offers a unique method for combining parameters, up to the choice of placing the first piece $a_0$.*

*Proof.* We can assume the dimension of target filter $d$ is even, as the odd case can be achieved by merging the middle rows and columns of the even case.

Our proof is structured as follows:

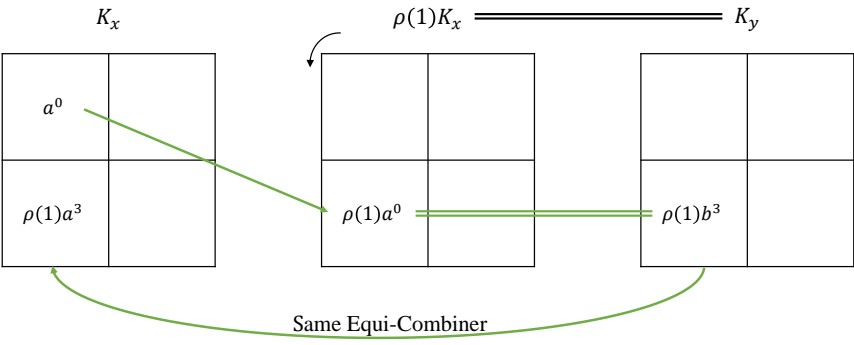

Figure 6: If the 0-th piece is assigned to the top left, the assignment of the bottom left corner is fixed.

Since $Z_4$ is a cyclic group generated by 1, it suffice to prove the case of $\rho(1)$, the 90-degree rotation.

Given an input image $x^0$, let $y^0 = \rho(1)x^0 = x^1$ be the 90-degree rotated version of $x^0$. For each input $x^0$ and $y^0$, we want to generate a filter for each, denoted by $K_x$ and $K_y$. Our assumption is $K_y = \rho(1)K_x$.

For $x^0$, we denote all rotated version of input by $x^1, x^2$ and $x^3$. Similarly, we denote rotated version of $y^0$ as $y^i$ for $i = 1, 2, 3$. Since $y^0 = x^1$, we know $y^{i+1} = x^i$ for all $i = 1, 2, 3$.

Given NEP generator $N$, denoted the output $a^i = N(x^i)$ and $b^i = N(y^i)$. Due to the relationship between $x$ and $y$, we also have $b^i = a^{i+1}$.

Now for the equi-combiner $E$, recall that it assigns the parameter pieces $a^i$ and $b^i$ to the filter $K_x$ and $K_y$ accordingly, based on the index. Assume $E$ assigns the 0-th parameter pieces on the top left corner. Therefore, $a^0$ is on the top left of $K_x$. By the assumption $K_y = \rho(1)K_x$, we know that the bottom left corner is $\rho(1)a^0 = \rho(1)b^3$. Since $E$ performs the same action based on index regardless of $a$ and $b$, we know that $\rho(1)a^3$ goes to the bottom left corner as well. The process is visualized in 6.

If we keep repeating this process, we get that $\rho(2)a^2$ goes to bottom left and $\rho(3)a^1$ goes to the top right corner. This is exactly the equi-combiner that we described, finishing the proof.

$\square$

**Proposition 8.** *Any filters with properties in proposition 7 grant equivariance to the convolutional main network $f$. That is, let $w$ be a hypernetwork and $\rho$ be the 90-degree rotation representation. If $w(\rho x) = \rho w(x)$, we have*

$$\rho f(x) = f_{w(\rho x)}(\rho x).$$

*Proof.* This simply follows from the fact that the element-wise multiplication of two equal-sized square matrices remains unchanged when both matrices are subjected to 90-degree rotations. $\square$

For linear case, we have a similar proposition, and the proof is fairly similar.

**Proposition 9.** *Let $\{c_i\}_{\{0,1,2,3\}}$ be the generated column vectors mentioned in Section 4.3. We have the following properties:*

1. *Our equi-combiner guarantees filter rotation as the inputs rotate.*

2. *The order of combining is unique up to the choice of placing the first piece $c_0$.*

3. *Let $\rho_{lin}$ be the representation in Section 4.3. If $f$ is a linear layer with input vector $v$, we have*

$$f(v) = f_{w(\rho_{lin}v)}(\rho_{lin}v).$$

*Proof.* The proof of the first and second statement is similar to the case of convolutional filters. For an input $x^0$ and a second input $y^0 = \rho(1)x^0$, we track the assignment of their generated column vectors.

The third statement directly follows from adding the same bias to both side of Eq. (2). $\square$

Finally, we show our results hold for additional layers. Given two square matrices, representing inputs and features in the full network $f_w$, we have the following proposition:

**Proposition 10.** *Given a square matrix $M$, we denote $M' = \rho M$ as the 90° rotation version of $M$.*

1. *For activation layers $A$ that performs entry-level computation operation such as ReLu and Softmax, the equivariance is preserved. That is, $A(M') = \rho A(M)$.*

2. *For pooling layers $P$ with square pooling filter size, the equivariance is preserved. That is, $P(M') = \rho P(M)$*

3. *Assume $M$ and $M'$ are matrices in batch. For Batch normalization layers $B$, the equivariance is preserved on the batch-level. That is, $B(M') = \rho B(M)$ if and only if $\rho$ is applied to all elements in batch.*

*Proof.* For the first statement, since all computation happens on the entry level, the rotation operation commutes with the activation layer, and have $A(\rho M) = \rho A(M)$. For cases of Softmax, since the sum of $M$ and $M'$ is identical, $A(\rho M) = \rho A(M)$ still holds.

For the second statement, the pooling layer $A$ can be viewed as a special case of 8, and the result $P(M') = \rho P(M)$ follows.

For the last statement, it is clear when $\rho$ is applied on the whole batch, we have equivariance. However, if $\rho$ is only applied on one of the input, we do not have equivariance due to the operation of batch normalization. $\square$

