# OpenReview forum: "Hypernetwork-Based Equivariant CNNs"
_ICLR.cc/2025/Conference — Submitted to ICLR 2025_

### Official Review · Reviewer_JNi2 · 2024-10-27

**Soundness:** 3
**Presentation:** 3
**Contribution:** 2
**Rating:** 5
**Confidence:** 4

**Summary:**

Equivariant deep learning methods have seen a growing amount of research interest over recent years as model paradigm of choice for small datasets with known symmetries. However, these methods have been shown ot scale poorly with more data, as the computational overhead or symmetry-specific constraints impede the optimization of such networks. This paper proposes a novel GCNN architecture that attempts to address these constraints. Authors propose to do this by parameterizing the weights of their equivariant model through a non-equivariant hypernetwork. Equivariance is achieved by augmenting the input data according to the group of choice, passing the data through this hypernetwork and composing the resulting output kernels and linear layers according to the group action. The resulting parameterized network, which the authors show to be equivariant by construction, is then used to e.g. classify the input data. Authors propose a number of methods to reduce the computational overhead of this approach, and ultimately show experimental results on a number of vision classification benchmarks - showing better or competitive performance to previous equivariant methods.

**Strengths:**

- The paper is well-written (save for some minor spelling/grammar mistakes), making the matter easy to understand. The complex concept of equivariance in deep learning is introduced in a very accessible manner.
- The proposed method is innovative, as it allows for using any non-equivariant base model to achieve an equivariant framework. This approach is different from canonicalization-based methods, as it retains its equivariance property throughout the network, arguably a desirable property in complex vision tasks with multiple objects/object-parts under different relative poses.
- The model's performance in experiments is strong compared to other equivariant deep learning baselines, achieving better or competitive performance in classification on CIFAR10/100 and STL10.
- The paper contains illustrative figures that aid understanding of the work.

**Weaknesses:**

- My main concern is with respect to the computational overhead that your method imposes. Every input sample needs to be passed through the NEP $|G|$ times, and as you yourself indicate, the resulting network parameters are sample-specific i.e. batched application of the same resulting kernel is not possible without averaging the sample-specific parameters over the batch. It would be good to indicate how your method compares in terms of memory and computational efficiency to base CNN and G-CNN models. It seems to me this computational requirement might be prohibitive in applying your model to larger-scale data (e.g. imagenet which has 224x224 resolution). Could you comment on this?
- The experimental validation of this method is somewhat limited. Especially the results in Sec. 5.1 are hard to valuate, the authors seem to intentionally have chose a very weak baseline (96% on MNIST can easily be achieved by simple MLPs), but i'm not sure why they chose this base model. It makes the results in this table very hard to judge on how informative they are. I think it would be worthwhile to additionally include results for non-invariant tasks (e.g. segmentation).
- If you're okay with performing $|G|$ forward passes per sample, why not simply apply a non-equivariant model directly to the augmented samples and treating the stacked outputs as an equivariant feature map, possibly performing a group-invariant projection over the results? This would also yield a model equivariant/invariant to the group action and is arguably simpler, no?
- Your main motivation is to lift the constraints that equivariance imposes on model architectures and their optimization. However, isn't your hypernetwork by construction also constrained by the same equivariance constraints that G-CNN filter kernels have? In G-CNNs kernels are constructed by transforming a "canonical" kernel under $g\in G$. This construction leads to specific constraints on the optimization: any gradient update is invariant to transformations of an image with $g\in G$. Aren't your kernels constructed in much the same way, leading to the same constraints on gradient updates as in the original G-CNN formulation? What then would be the reason for improved performance of your method compared to G-CNN baselines?

**Questions:**

- Could you please expand on how your work relates and contrasts from the work of Zhdanov et al [1], who also use a hypernetwork-based approach to parameterize steerable convolutional kernels? I think it might be good to mention this work.
- Could you provide details on the computational complexity of your method, i.e. its runtime and memory efficiency compared to the base CNN and G-CNN models? I think this is an important factor to consider in the evaluation of your work, as it might help the reader gauge how well your method would scale to larger-scale data.


[1] Zhdanov, M., Hoffmann, N., & Cesa, G. (2024). Implicit convolutional kernels for steerable CNNs. Advances in Neural Information Processing Systems, 36.

---

> ### Author Response · Authors · 2024-11-25
>
> Thank you for your constructive comments. Below we have made responses to your comments. If you have any further comments, please feel free to let us know and we are more than glad to discuss with you.
>
> >Question 1: Could you please expand on how your work relates and contrasts from the work of Zhdanov et al[5], who also use a hypernetwork-based approach to parameterize steerable convolutional kernels? I think it might be good to mention this work.
>
> We thank you for referencing this paper. In this work by Zhdanov et al. [r1], they used a MLP to generate $G$-Steerable kernels, instead of computing kernel basis for each particular group $G$. This definitely falls into the category of a hypernetwork. While their work simplifies the process of finding steerable kernels, the main difference from our approach is that their main network is still steerable CNNs. In contrast, our HE-CNN uses a regular CNN as our main network, and does not incorporate any G-CNN based structures. This related work will be added to the final version of our paper.
>
> [r1] Zhdanov, M., Hoffmann, N., & Cesa, G. (2024). Implicit convolutional kernels for steerable CNNs. Advances in Neural Information Processing Systems, 36.
>
> > Question 2: Could you provide details on the computational complexity of your method, i.e. its runtime and memory efficiency compared to the base CNN and G-CNN models? I think this is an important factor to consider in the evaluation of your work, as it might help the reader gauge how well your method would scale to larger-scale data.
>
> In the following table, we provide our model's training time per iteration comparing to base CNN and G-CNN on the CIFAR10 dataset based on the 13-layer CNN. We further trained G-CNN with a dynamic hypernetwork for comparison.
>
> | Model Trained with CIFAR10 | Test Accuracy (%) | Parameters | Training Time per iteration |
> | --- | ---: | ---: | ---: |
> | 13-Layer CNN | 91.21 | 3,119,754 | 18.67s/it |
> | Z4-CNN | 89.73 | 12,456,586 | 87.92s/it |
> | Z4-HE-CNN | 91.95 | 380,271,050 | 151.77s/it |
> | Z4-HE-CNN (LoRA+Head) | 90.12 | 23,129,268 | 139.82s/it |
> | Z4-CNN trained with Dynamic Hypernet w/ LoRA | 89.84 | 32,627,215 | 147.21s/it |
>
>
> >Weakness 1: My main concern is with respect to the computational overhead that your method imposes, passing inputs through the NEP $|G|$ times. It would be good to indicate how your method compares in terms of memory and computational efficiency to base CNN and G-CNN models. It seems to me this computational requirement might be prohibitive in applying your model to larger-scale data (e.g. imagenet which has 224x224 resolution). Could you comment on this?
>
> We agree with you that our current limitations on scalability, primarily due to the usage of hypernetworks and generating input-dependent parameters. Our current solution utilized LoRA to reduce the parameter of the last linear layer in the NEP generator. The comparison of parameter count and training time are detailed in our response to Question 2. The reason for not applying our method to larger-scale datasets is that there are few applications for baseline equivariant CNNs on ImageNet/Tiny ImageNet. As of this response, we found no reports based on G-CNN or steerable CNNs on the Paper with Code website for ImageNet/Tiny ImageNet. We choose Rotated MNIST, STL10 and CIFAR10/100 to match datasets chosen in group equivariant CNNs [r2] and steerable CNNS [r3,r4].
>
> [r2] Cohen, T., & Welling, M. (2016, June). Group equivariant convolutional networks. In International conference on machine learning (pp. 2990-2999). PMLR.
> [r3] Cohen, T. S., & Welling, M. (2016). Steerable cnns. arXiv preprint arXiv:1612.08498.
> [r4] Weiler, M., & Cesa, G. (2019). General e (2)-equivariant steerable cnns. Advances in neural information processing systems, 32.
>
> > Weakness 2: The experimental validation of this method is somewhat limited. Especially the results in Sec. 5.1 are hard to valuate, the authors seem to intentionally have chose a very weak baseline (96% on MNIST can easily be achieved by simple MLPs), but i'm not sure why they chose this base model. It makes the results in this table very hard to judge on how informative they are. I think it would be worthwhile to additionally include results for non-invariant tasks (e.g. segmentation).
>
> We appreciate your concern. A more detailed comparison of our methods on Rotated MNIST is done in section 5.2, utilizing a significantly larger models. As of section 5.1, such experiment are designed not to show the full capacity of each methods. Rather, we want to empirically show that constraints on the $Z4$-filters negatively impact performances, especially when trained and tested on the original MNIST. We believe a smaller model would effectively highlights these constraints, and we ensured that the network sizes are consistent across all setups for a fair comparison.

---

> > ### Author Response · Authors · 2024-11-25
> > **Continuation of Prior Comments**
> >
> > > Weakness 3: If you're okay with performing $|G|$ forward passes per sample, why not simply apply a non-equivariant model directly to the augmented samples and treating the stacked outputs as an equivariant feature map, possibly performing a group-invariant projection over the results? This would also yield a model equivariant/invariant to the group action and is arguably simpler, no?
> >
> > We thank the reviewer for suggesting the comparison of an additional approach. We extend our experiments in section 5.1 with two additional approach with similar model size as an ablation study. The"CNN + All $90^\circ$ rotated input" model sums the outputs of all rotated versions of inputs as the final output. It is trained on the training set of the Rotated MNSIT, and tested on all three testing datasets. Performing |G| forward passes directly to a non-equivariant models is used in equi-tuning [r5]. It has demonstrated its effectiveness in the context of large models and fine-tuning. In our case, our experiment showed that it grants equivariance to the model, but resulted in little improvement over CNN and $Z4$-CNN. This method requires each filter to perform well on all rotated versions of inputs, imposing a similar constraint with $Z4$-CNN where all rotated versions of filters are required to perform well on inputs. Additionally, we also included dynamic $Z4$-CNN model in the ablation study, utilizing a dynamic hypernetwork to train a $Z4$-CNN, as suggested by reviewer m86f.
> >
> >
> > |          Performance (%) of models trained on Randomly Rot-MNIST             | CNN   | $Z4$-CNN | Ours  | Dynamic $Z4$-CNN | CNN + All $90^\circ$rotated input |
> > | :------------------------ | :---- | :------- | :---- | :--------------- | :-------------------------------- |
> > | Original MNIST            | 75.27 | 76.22    | 96.44 | 76.19            | 75.38                             |
> > | Randomly Rot MNIST             | 73.08 | 75.51    | 96.32 | 74.54            | 75.21                             |
> > | $90^\circ$  rotated MNIST | 75.68 | 76.22    | 96.44 | 76.03            | 75.38                             |
> > | *Parameter*    | *3370*  | *12522*    | *11527* | *32964*            | *3370*                              |
> >
> > [r5] Basu, S., Sattigeri, P., Ramamurthy, K. N., Chenthamarakshan, V., Varshney, K. R., Varshney, L. R., & Das, P. Equi-tuning: Group equivariant fine-tuning of pretrained models. In Proceedings of the AAAI Conference on Artificial Intelligence, 2023.
> >
> > > Weakness 4: Your main motivation is to lift the constraints that equivariance imposes on model architectures and their optimization. However, isn't your hypernetwork by construction also constrained by the same equivariance constraints that G-CNN filter kernels have? In G-CNNs kernels are constructed by transforming a "canonical" kernel under $g\in G$. This construction leads to specific constraints on the optimization: any gradient update is invariant to transformations of an image with $g\in G$. Aren't your kernels constructed in much the same way, leading to the same constraints on gradient updates as in the original G-CNN formulation? What then would be the reason for improved performance of your method compared to G-CNN baselines?
> >
> > We would like to clarify that the constraint we discussed in our paper are referring to a different concept as the constraints you mentioned. As I understand it, constraints that you mentioned refer to gradient updates being equivariant to group actions. However, we believe that $G$-CNN's effectiveness depends on such equivariance. If gradient updates are completely different for $x$ and $gx$, the model cannot be equivariant to $G$.
> > The constraint we mentioned is different. Given a "canonical" kernel $K$, G-CNN requires all rotated versions of $K$ to behave well with the inputs. Optimizing all rotated versions simultaneously imposes constraints on the gradients update. However, our $HE$-CNN only needs to optimize the generated $K$ for the input $x$. Unlike $G$-CNN, rotated versions of $K$ are not required to work well with $x$.

---

> > ### Comment · Reviewer_JNi2 · 2024-11-26
> >
> > I appreciate the response by the authors. I do agree with the authors that e.g. imagenet is not classically used in equivariance literature, precisely because of scaling problems. I cannot fault the authors for considering this out of scope.
> >
> > I thank the authors for providing details regarding computational complexity of their method compared to the baselines. I am quite surprised by the comparison made by the authors; it seems to me that a 380M model is excessive for CIFAR (e.g. ResNet18 generally has around 270K parameters achieving 92% accuracy on CIFAR), and given that the CNN baseline is 3M params and almost 100x more time efficient, which to me makes interpreting these results very hard to do meaningfully. What do the authors think of these comparisons?
> >
> > You mention that in Sec 5.1, the goal of using smaller methods is to compare the impact of the different constraints on each model, but how can you be sure that the conclusions you draw from these results generalize to models of practical size? Als you mention "we ensured that the network sizes are consistent across all setups", but it seems from Tab 2. that the CNN model you are comparing against has 3K parameters, whereas your model has 12K parameters? Or am I misunderstanding?

---

> > > ### Author Response · Authors · 2024-12-02
> > >
> > > We thank the reviewer for your constructive replies. Below, we provide responses to your feedback. If you still have any questions or need further clarification, please feel free to let us know, and we would be happy to clarify.
> > >
> > > > Question 1: it seems to me that a 380M model is excessive for CIFAR (e.g. ResNet18 generally has around 270K parameters achieving 92% accuracy on CIFAR), and given that the CNN baseline is 3M params and almost 100x more time efficient, which to me makes interpreting these results very hard to do meaningfully. What do the authors think of these comparisons?
> > >
> > > We include $Z4$-HE-CNN to demonstrate the maximum potential of our model, which features a large final linear layer. The model Z4-HE-CNN (LoRA+Head) provides a more practical comparison with $Z4$-CNN, with 1.59 times of training time and approximately twice the parameter counts.
> > > While both our method and $G$-CNN are less efficient and have more parameters, the advantage of equivariance grants rotational robustness to our model. Given that CIFAR10 has relatively small rotation variations compared to Rot-MNIST, this equivariance offers limited benefit in test accuracy, particularly when the group $G$ is small.
> > > As you noted, the base CNN is indeed powerful and efficient. However, it generally underperforms when presented with rotations due to the lack of equivariance properties, as demonstrated in Section 5.2 on the Rot-MNIST dataset. Since our goal is to design alternative equivariant models, our primary comparison is with G-CNN.
> > > We respectfully believe that ResNet18 is generally reported to have approximately 11 million parameters, rather than the 270k mentioned in the review. This parameter is counted using the default ResNet18 implementation in torchvision.models. We hope this clarification allows for a more accurate and fair comparison of our models.
> > >
> > >
> > >
> > > > Question 2: For Tab 2 in Section 5.1,  the goal of using smaller methods is to compare the impact of the different constraints on each model, but how can you be sure that the conclusions you draw from these results generalize to models of practical size? Also you mention "we ensured that the network sizes are consistent across all setups", but it seems from Tab 2. that the CNN model you are comparing against has 3K parameters, whereas your model has 12K parameters? Or am I misunderstanding?
> > >
> > > To address the first part of your question, we chose a small network to empirically demonstrate the effects of the constraints. As the network size increases, these effects diminish. However, theoretically, simultaneously optimizing a single kernel for all rotated images introduces constraints on the gradients. This results in regular CNN architectures often achieving higher maximum potential performance than G-CNN structures on various benchmark datasets with relatively small rotation variations, such as CIFAR-10/100 and STL-10. Consequently, current group equivariant structures are typically employed in tasks that are closely related to symmetries.
> > >
> > > To answer the second part of your question, when we referred to "sizes", we were specifically discussing the number of hidden layers, hidden channels, and filter sizes. These aspects are consistent between the $Z4$-CNN and base CNN. Due to the design of $Z4$-Convolution, their parameters are approximately four times the parameter count of the base CNN. Our hypernetwork is designed to have similar parameter counts as $Z4$-CNN, while our main network (the network that processes inputs using the generated parameters) remains consistent with the base CNN architecture.
> > >
> > > However, we acknowledge that our previous statement may have caused potential confusion. We will revise that sentence to read: "We ensured that the network's hidden channels and filter sizes are consistent between the $Z4$-CNN, CNN and the main network of our approach. Additionally, our hypernetwork is designed to have a similar parameter count to the $Z4$-CNN."

---

### Official Review · Reviewer_kvp3 · 2024-11-01

**Soundness:** 2
**Presentation:** 2
**Contribution:** 2
**Rating:** 5
**Confidence:** 3

**Summary:**

This paper introduces Hypernetwork-based Equivariant CNNs (HE-CNN), a hypernetwork-based approach to achieve rotational equivariance in convolutional neural networks. Unlike Group Equivariant CNNs (G-CNNs) that impose a filter constraint, HE-CNN uses a dynamic hypernetwork to generate input-dependent parameters that inherently exhibit equivariance. The hypernetwork consists of a non-equivariant parameter-pieces generator and an equivariant combiner that assembles the pieces to form equivariant parameters. The authors provide theoretical proofs of the model's equivariance properties and demonstrate its effectiveness through experiments on multiple benchmark datasets including MNIST, CIFAR10/100, and STL10. HE-CNN demonstrates comparable performance on those datasets.

**Strengths:**

- The authors propose to directly utilize the dynamic hypernetwork to generate equivariant parameters.
- The proposed design of using hypernetwork for equivariance is validated in small-scale experiments.

**Weaknesses:**

- The 90-degree rotation settings is over-simplified to some extent. It is easier for the hypernetwork to learn the G-equivariant parameters compared with a larger rotation group. Especially in the linear layer setting, if the rotation degree is not in {0, 90, 180, 270}, the rotated feature cannot be easily represented as permutation which is the premise of the hypernetwork design.
- The current presentation, especially equations and figures, is a bit hard to interpret. For example, in Line 204, what’s the dimension of the output $a^i$.
- The proposed method does not show competitive results compared with the given limited benchmarks, especially in experiments with deeper CNNs on the CIFAR-10/100 datasets.

**Questions:**

-Could you please elaborate on the detailed hypernetwork design used in the experiments?

---

> ### Author Response · Authors · 2024-11-25
>
> Thank you for your constructive comments. Below we have made responses to your comments. If you have any further comments, please feel free to let us know and we are more than glad to discuss with you.
>
> >Weakness 1: The 90-degree rotation settings is over-simplified to some extent. It is easier for the hypernetwork to learn the G-equivariant parameters compared with a larger rotation group. Especially in the linear layer setting, if the rotation degree is not in {0, 90, 180, 270}, the rotated feature cannot be easily represented as permutation which is the premise of the hypernetwork design.
>
> We appreciate your insight regarding our HE-CNN, which currently supports a subset of common transformations. In Section 4.5, we extend the 90-degree rotation to rotations of 90/n degrees. Similar to the G-CNN, we are currently limited to translation and discrete rotations. However, we are optimistic that these limitations can be addressed in future work. As the E(2) steerable CNN [r1] extends the G-CNN to the continuous case, we look forward to learning from it to enhance our HE-CNN for future exploration.
>
> [r1] Weiler, M., & Cesa, G. (2019). General e (2)-equivariant steerable cnns. Advances in neural information processing systems, 32.
>
> >Weakness 2: The current presentation, especially equations and figures, is a bit hard to interpret. For example, in Line 204, what’s the dimension of the output $a^i$.
>
> Thank you for pointing out potential sources of confusion. In Figure 2, we intended to show that each $a^i$ is a quarter of the full filter. When the full filter is in dimension $K$, the exact dimension of each $a^i$ is (out channel, in channel, $\lceil \frac{K}{2} \rceil$, $\lceil \frac{K}{2} \rceil$), as mentioned in line 198 on page 4. We agree that we should make the connection more clear, and adjustments will be made in the final version.
>
> >Weakness 3: The proposed method does not show competitive results compared with the given limited benchmarks, especially in experiments with deeper CNNs on the CIFAR-10/100 datasets.
>
> We appreciate your concerns  regarding the comparison of our method with researches building upon G-CNN. We would like to clarify that our HE-CNN aims to achieve equivariance through a novel approach while reducing the constraints present in G-CNN-based structures. Thus, main baseline methods in comparison are G-CNN-based methods such as partial equivariance [r2], which are largely derived from G-CNN. Since steerable CNNs, along with most equivariance structures in image processing, are based on G-CNN, we believe our novel alternative has significant potential for future extensions.
>
> [r2] Romero, D. W., & Lohit, S. (2021). Learning partial equivariances from data. arXiv preprint arXiv:2110.10211.
>
> >Question 1: Could you please elaborate on the detailed hypernetwork design used in the experiments?
>
> For a general guide line, our hypernetwork should output approximately a fourth of the main network's parameters. For our implementation, All our Hypernetwork are CNN based. For experiment in section 5.1, our NEP generator is composed of 2 hidden convolutional layers with 16 hidden channels and three by three kernels. We have one linear head for each layer in the main network, and used LoRA for each layer with intermediate rank of 4. For experiment in section 5.2, the NEP generator is composed of a 3-layer convolution with [16,32,32] intermediate channels,[3,3,4]filter size and [2,2,1] stride. If LoRA is utilized, the intermediate rank is set to 4. For the experiment on CIFAR10/100, the NEP generator has 3 convolution layer with [32,64,64] channels with all filter sizes 3. Batch normalization, relu and max-pooling layers are inserted between convolutional layers. Stride is set to 2 for the last convolutional layer. When LoRA is used, intermediate rank is set as 7. For STL10, NEP generator is composed of 4 convolutional layers with [64,128,128,64] intermediate channels with filter size 3. Batch normalization, relu and max-pooling layers are inserted between convolutional layers. Pooling are down after first three convolutional layers.

---

> > ### Comment · Reviewer_kvp3 · 2024-11-26
> >
> > Thank authors for their responses. However, the authors’ responses do not fully address my concerns, especially regarding the first weaknesses of the review. Although the authors show how to extend the HE-CNN to $D_{4n}$, it dramatically increases the computation cost and memory cost by forwarding both the hypernetwork and the main CNN for $|D_{4n}/Z_4|$ times. Moreover, this extension reduces the technical novelty to some extent. As we can already obtain equivariant features by feeding multiple rotated input images to the static CNN, why would we still require a complex hypernetwork structure? Besides, I agree with other reviewers that the current submission adopts outdated baselines, making the experimental validation unconvincing, especially in the CIFAR10/100 experiments. Lastly, the authors haven’t provided a revision for a better presentation. The current presentation needs significant refinement in figures and notations. Given the reasons above, I will maintain my score.

---

> > > ### Author Response · Authors · 2024-12-02
> > >
> > > We thank the reviewer for the response. Below we have made responses to your comments.
> > > > As we can already obtain equivariant features by feeding multiple rotated input images to the static CNN, why would we still require a complex hypernetwork structure?
> > >
> > > While it is true that we can obtain equivariant features by feeding all rotated versions of the input to a base CNN, our ablation study revealed that this approach yields only a limited performance improvement over the base model.
> > >
> > > |          Performance (%) of models trained on Randomly Rot-MNIST             | CNN   | $Z4$-CNN | Ours  | Dynamic $Z4$-CNN | CNN + All $90^\circ$rotated input |
> > > | :-- | :--- | :------- | :---- | :---------- | :-------- |
> > > | Original MNIST            | 75.27 | 76.22    | 96.44 | 76.19            | 75.38                             |
> > > | Rotated MNIST             | 73.08 | 75.51    | 96.32 | 74.54            | 75.21                             |
> > > | $90^\circ$  rotated MNIST | 75.68 | 76.22    | 96.44 | 76.03            | 75.38                             |
> > > | Parameter    | 3370  | 12522    | 11527 | 32964            | 3370                              |
> > >
> > > > The current presentation needs significant refinement in figures and notations.
> > >
> > > We apologize for any confusion we may have caused. Could you please specify which figure or notation seems unclear? We would be more than happy to provide clarification.

---

### Official Review · Reviewer_m86f · 2024-11-03

**Soundness:** 2
**Presentation:** 2
**Contribution:** 2
**Rating:** 5
**Confidence:** 4

**Summary:**

This work presents HE-CNNs. An equivariant neural architecture method, where the network parameters are generated based on a hypernetwork in a way that it is equivariant to a defined (finite & small) group. The authors compare their method to G-CNNs and demonstrate competitive results with regard to existing state-of-the-art (although relatively old) methods.

**Strengths:**

- To the best of my knowledge, the way proposed in this paper to construct equivariant filters is novel. I’ve not seen such an approach before.

**Weaknesses:**

While the method proposed in this paper is new, I have concerns along several different axes: scalability, the conclusions drawn, and the analysis of why the method works.

- My most important concern is that the scalability of the method is really problematic. This touches upon compute efficiency, large networks, large batches, large groups, etc. Note that some of these issues are not even mentioned in the paper. The paper lacks an explicit limitations section as well, where all of these scaling issues should be clearly stated.

  - First, in terms of computation, this method basically uses a neural network to generate the weights of another neural network. Consequently, one could argue that this imposes important constraints on the size that the hypernetwork may have. Unfortunately, the authors do not touch upon runtime etc compared to non-hypernetwork-based networks.
  - Next, the method requires passing N versions of the input through the network –where N is the size of the group– for every single input in an input batch, in order to generate equivariant weights. This becomes a clear constraint for large –let alone continuous– groups. The authors also do not mention this.
  - Then, the hypernetwork must generate weights for the whole of the other network. This is another important scalability constraint over the size of the hypernetwork required in those settings. It also calls into question how big that network needs to become for growing sizes of the main network. The authors use LoRA to this end, but in my view this does not solve the scalability issues of the method.
  - Next, the authors mention that the method is equivariant only if all inputs in a single batch have the same orientation. This is, of course, very unrealistic and in my view an extremely constraining limitation. As an alternative, one can generate different network weights for each one input, but this, again, does not scale.

- Aside from the scalability issue, I am also concerned about the last point mentioned above. From what I understand, the method assumes that it is known what g transformation is applied to the input. Note that, if this were the case in realistic settings, this would dramatically simplify how equivariant methods work.

- Next, the authors argue that the reason HE-CNNs work better than baseline G-CNNs is the fact that the kernels in G-CNNs are constrained. However, I am inclined to believe that there are, in fact, two composing factors leading to better results. Note that, in contrast to G-CNNs, the weights of HE-CNNs are input dependent. Yet, there is no analysis on whether the gains result from not constraining kernels in the main input, or from this input dependency. I emphasize that lack of this analysis may be leading to incorrect conclusions. For instance, it could be completely possible that the gains are not from constraining the kernels, but in fact because these kernels are input dependent. I encourage the authors to include such analyses.

**Questions:**

* Is it correct that the method assumes that the orientation of the input is know beforehand? If not, how is it that  the $a_r^i$ can be described relative to a canonical direction?

### Limitations

The authors do not state clearly the limitations of their method. There is no specific section dedicated to this.

### Conclusion

While I believe that the method proposed here is novel, I have several concerns about the method proposed, its inner working and its use in practice. I believe there is potential, but, in my view, the method, the paper and the experimental analysis must be polished before this paper can be accepted. I am therefore, unable to support acceptance at this point.

---

> ### Author Response · Authors · 2024-11-25
>
> Thank you for your constructive comments. Below we have made responses to your comments. If you have any further comments, please feel free to let us know and we are more than glad to discuss with you.
>
> >Question 1: Is it correct that the method assumes that the orientation of the input is know beforehand? If not, how is it that the $a^i$ can be described relative to a canonical direction?
>
> We apologize for the confusion and thank you for the opportunity to clarify. We do not assume the orientation of the inputs or the group action is known beforehand. What we meant at the end of section 4.6 is that, due to the fact that we take the average approach when generating parameters, equivariance in our method is satisfied if and only if the group action is applied onto all images in the batch. We would like to emphasize that this relaxation of the definition of equivariance is also true for G-CNN when batch normalization is applied within the layers.
>
> To answer your question about $\{a^i\}$: For an input $x$, we collect all rotated versions of $x$. This entire collection is sent to the NEP generator, resulting in all $\{a^i\}$. Note that input $x$ can be in any orientation, not necessarily the canonical position. Our equi-combiner ensures that the final generated filters follow any rotation of the inputs, regardless of the initial position of $x$.
>
> >Weakness 1: The scalability of the method is really problematic.
> >
> >1a. In terms of computation, this method basically uses a neural network to generate the weights of another neural network. Consequently, one could argue that this imposes important constraints on the size that the hypernetwork may have.
> >
> >1b. The method requires passing N versions of the input through the network, becoming a clear constraint for large groups.
> >
> >1c. The hypernetwork must generate weights for the whole of the other network, purposing scalability issues of the method.
>
>
> Weakness 1a+1c: In the table below, we provide a comparison of the number of parameters and training time of the 13-layer CNN used on the CIFAR10 dataset. We also trained the Z4-CNN with a similar dynamic hypernetwork for comparison. In the revision, we will include a dedicated section discussing our current limitations in depth.
>
> | Model Trained with CIFAR10 | Test Accuracy(%) | Parameters | Training Time per iteration |
> | --- | ---: | ---: | ---: |
> | 13-Layer CNN (Base) | 91.21 | 3,119,754 | 18.67s/it |
> | Z4-CNN | 89.73 | 12,456,586 | 87.92s/it |
> | Z4-HE-CNN | 91.95 | 380,271,050 | 151.77s/it |
> | Z4-HE-CNN (LoRA) | 90.12 | 23,129,268 | 139.82s/it |
> | Z4-CNN trained with Dynamic Hypernet (LoRA) | 89.84 | 32,627,215 | 147.21s/it |
>
>
> Weakness 1b. We agree that passing inputs $N$ times is currently a limiting factor, particularly when $N$ is large. The G-CNN also faces this limitation, required to process one input with N rotated filters. The Steerable CNN [r1] helps mitigate this limitation, and E(2)-Equivariant Steerable CNNs [r2] extends basic G-CNN to continuous settings. Therefore, we believe in the potential of HE-CNN to overcome this limitation in future work.
>
> [r1] Cohen, T. S., & Welling, M. (2016). Steerable cnns. arXiv preprint arXiv:1612.08498.
> [r2] Weiler, M., & Cesa, G. (2019). General e (2)-equivariant steerable cnns. Advances in neural information processing systems, 32.
>
> > Weakness 1d. The authors mention that the method is equivariant only if all inputs in a single batch have the same orientation. This is very unrealistic and an extremely constraining limitation.
> > Weakness 2: Similar to weakness 1d, from what I understand, the method assumes that it is known what g transformation is applied to the input.
>
> Weakness 1d and 2: As mentioned in the response to Question 1, we do not assume that all inputs are presented in the same orientation, and we do not require information about the exact group action applied to the inputs.

---

> > ### Author Response · Authors · 2024-11-25
> > **Continuation of Prior Comments**
> >
> > > Weakness 3: There are two composing factors leading to better results. The weights of HE-CNNs are input dependent. Yet, there is no analysis on whether the gains result from this input dependency.
> >
> > We thank the reviewer for pointing this out. In the following table, we provide an ablation study. We maintain the same settings as in Section 5.1, testing small CNNs, G-CNNs, and HE-CNNs on different MNIST datasets. We included an additional model in the ablation study, utilizing a dynamic hypernetwork to train the G-CNN. The dynamic hypernetwork is chosen to be the same as that used in HE-CNN. LoRA (with the same rank) and multi-head is also used for a fair comparison. As shown in the table, dynamic hypernetwork did not result in a significant improvement in the test accuracy of the G-CNN. Additionally, we added ablation study suggested by reviewer JNi2, and feed all rotated versions of input to a non-equivariant CNN and average the output. This ablation study empirically demonstrates that our improvements are not due to the utilization of all rotated versions of the inputs.
> >
> > |  Performance (%) of models trained on Randomly Rot-MNIST  | CNN | $Z4$-CNN | Our | Dynamic $Z4$-CNN | CNN + All $90^\circ$rotated input |
> > | :--- | :---- | :--- | :---- | :---- | :---- |
> > | Original MNIST       | 75.27 | 76.22    | 96.44 | 76.19 | 75.38  |
> > | Rotated MNIST        | 73.08 | 75.51    | 96.32 | 74.54   | 75.21       |
> > | $90^\circ$  rotated MNIST | 75.68 | 76.22    | 96.44 | 76.03    | 75.38 |
> > | Parameter   | 3370  | 12522    | 11527 | 32964            | 3370  |

---

> > ### Comment · Reviewer_m86f · 2024-11-25
> >
> > Dear authors,
> >
> > Thank you for your clarifications. I am looking forward to see the updated version of the manuscript.
> >
> > * Our equi-combiner ensures that the final generated filters follow any rotation of the inputs, regardless of the initial position of $x$.
> > > I am sorry, but I still cannot see how this works. Would you mind giving a brief explanation of how this is achieved?
> >
> > * regarding this comment:  "We would like to emphasize that this relaxation of the definition of equivariance is also true for G-CNN when batch normalization is applied within the layers.".
> > > This is not the case. When using batch norm, the statistics are handled across the channels. If you look at existing G-CNN implementations --I wont link to any particular one here for the sake of the double blind process--, most of them use e.g., BatchNorm3d when working with images batched as $[B, N_h, W, H]$. The reason of this is to account for the fact that group representations will have an additional dimension $G$, e.g., for rotations, leading to a latent representation $[B, N_h, G, W, H]$. What you indicate would be true if the shape used for batchnorm were $[B, N_h * G, W, H]$. But exactly to avoid this problem, this is implemented in a different way.

---

> > > ### Author Response · Authors · 2024-12-02
> > >
> > > We thank the reviewer m86f for their constructive comments. Below, we provide responses to your feedback. If you still have any questions or need further clarification, please feel free to let us know, and we would be happy to clarify.
> > >
> > > >Reply #1: The reviewer is still uncertain of how the equi-combiner generate parameters that rotate as input rotates.
> > > >
> > > We are happy to further clarify our parameter generation process. Let us first consider a simple case of generating one single 2x2 filters. We denote the NEP generator as $N$. In this case of generating 2x2 filters, the output dimension of $N$ is simply $[1,1]$.
> > >
> > > Given an input image $x^0$, our objective is to demonstrate that the generated filters are invariant to rotations of the image. Denote the $90^\circ$-rotated version of $x^0$ as $y^0=\rho(1)x^0$.  First, we will outline the process of our equi-combiner for $x^0$ and $y^0$, and then we will explain how the generated filters are related.
> > >
> > > Starting with $x^0$, we send all rotated versions, $\{x^0, x^1,x^2,x^3\}$, into $N$, resulting in $\{a^n|a^n=N(x^n)\}_{n\in Z_4}$. The table below illustrates how our equi-combiner assembles these pieces into a full filter $K_x$ in a counter-clockwise manner:
> > > |   $a^0$  |  $a^1$   |
> > > | --- | --- |
> > > |   $a^3$  |   $a^2$  |
> > >
> > > Now, we apply the same process for $y^0$. We denote the outputs as $\{b^n|b^n=N(y^n)\}_{n\in Z_4}$. The equi-combiner assembles $\{b^n\}_{n\in Z_4}$ in the same manner, resulting in a corresponding filter $K_y$:
> > > |   $b^0$  |  $b^1$   |
> > > | --- | --- |
> > > |   $b^3$  |   $b^2$  |
> > >
> > > Now, since we know that $y^0$ is a rotated version of $x^0$, we know that $y^n = x^{n+1}$ with the modulo 4 addition. Therefore, after the NEP generator $N$, we have $b^n = a^{n+1}$. Thus, we can rewrite $K_y$ as
> > >
> > > |   $a^1$  |  $a^2$   |
> > > | --- | --- |
> > > |   $a^0$  |   $a^3$  |
> > >
> > > , which is exactly the $90^\circ$ rotated version of $K_x$. This concludes the discussion for the 2x2 filter case. Note that no additional information is needed for $x^0$.
> > >
> > > Now, when the filters are larger than 2x2, each $a^i$ has dimension larger than $1$. Therefore, we must consider the rotation of each $a^i$ as well. Fortunately, this can be addressed through additional rotations, as mentioned in Section 4.2 in the paper. The equi-combiner uses the adjusted $\{a_r^i\}$, performing a **clockwise** $i\cdot 90^\circ$ rotation for each $a^i$. Our equi-combiner in the general case can be described using the table below
> > > |   $a_r^0$  |  $a_r^1$   |
> > > | --- | --- |
> > > |   $a_r^3$  |   $a_r^2$  |
> > >
> > > , as shown in Figure 2 on the bottom of page 4. Following a similar process as the 2x2 case, one can check that: for $x^0$ and $y^0=\rho(1)x^0$, their generated filters are also related through 90-degree rotation.
> > >
> > >
> > > >Reply #2: In the previous reply, the author made a mistake stating that G-CNN is equivariant only if the group action is applied on the whole batch of inputs when batch normalization is present.
> > >
> > >
> > > We would like to thank the reviewer for the insightful correction. During our testing, we indeed applied reshaping and 2D batch normalization to the outputs of a G-CNN layer. As the reviewer pointed out, while this approach leads to non-equivariance, many G-CNNs utilize 3D batch normalization to address this issue and maintain equivariance.
> > >
> > > In our paper, we did not make any misleading claims regarding this issue. However, we will ensure to include this discussion in the limitations section of the final version.

---

> > > > ### Comment · Reviewer_m86f · 2024-12-02
> > > >
> > > > Dear authors,
> > > >
> > > > I understand. Thank you very much for the clarification.
> > > >
> > > > Based on your clarifications, I have decided to increase my score to 5. The reason that I do not raise my score further is that, despite the clarifications, I still consider that there remain multiple pressing issues inherent to the method in its current form that prevent it from being truly impactful. It is not to say that I do not believe this method has potential, but rather that additional work is necessary to address these issues --particularly those related to scalability and unrealistic input assumptions--. I encourage the authors to keep developing the method proposed further.
> > > >
> > > > Best,
> > > >
> > > > Reviewer m86f

---

> > > > > ### Author Response · Authors · 2024-12-04
> > > > >
> > > > > We appreciate your decision to increase the score from 3 to 5; your constructive feedback is invaluable. We are committed to addressing the potential issues we discussed and will actively pursue solutions to enhance the quality of our work. We are grateful for your insights and support.

---

### Official Review · Reviewer_TBde · 2024-11-04

**Soundness:** 2
**Presentation:** 3
**Contribution:** 1
**Rating:** 5
**Confidence:** 4

**Summary:**

This paper proposes using hypernetworks to generate convolutional filters and coefficients based on rotated inputs. These filters are then applied in a convolutional layer, while the coefficients are used in a subsequent linear layer, enabling translation-rotation equivariance. Compared to the first G-CNN, the proposed method achieves higher accuracy, demonstrating its effectiveness even with minimal rotational variation.

**Strengths:**

The idea of using hypernetwork to achieve rotation equivariant filters and coefficients is interesting.

**Weaknesses:**

1. Outdated and Inconsistent Baselines: The proposed method is shown to perform better than the original G-CNN by Cohen & Welling (2016). However, recent advances in steerable G-CNNs already address rotational variation effectively. While this paper includes steerable G-CNN results for the rotated MNIST experiment, none are tested in the CIFAR-10/100 experiments. Additionally, the proposed method underperforms compared to E2CNN, a steerable G-CNN, raising questions about the strength of this baseline comparison. Given more advanced methods are available, outperforming only the original G-CNN is not sufficiently convincing. Moreover, the steerable G-CNN has a nice theoretical explanation, which is related to Nyquist–Shannon sampling theorem, whereas the proposed method is empirical.
2. Missing Parameter Counts: Matching the number of network parameters is crucial to ensure that performance gains are not simply due to an increase in parameters. Parameter counts are absent, making it difficult to assess the efficiency of the proposed method.
3. Questionable Results in Table 3: The results in Table 3 lack thorough discussion and seem questionable. P4-CNN and Z4-HE-CNN exhibit similar translation-rotation group equivariance on rotated MNIST. Assuming they have similar parameter counts, this could indicate that the proposed hypernetwork-based method may improve over equivariance-based methods for minimal rotational variation. However, models like LieConv, Steerable-CNN, E2FCNN, and Sim2-CNN are designed for continuous rotation and show optimal results with 16 orientation samples, yet they perform worse than the proposed Z4-HE-CNN. The results of Z4-HE-CNN (LoRA) appear more reasonable than Z4-HE-CNN. Without matched number of parameters, the state-of-the-art claim is questionable.

4. Limited Generalizability to Other Transformations: The use of hard-coded 90-degree rotations (Figure 2) and specific permutations (Figure 3) may limit generalizability to transformations like scaling and shear. Although 90-degree rotations avoid interpolation issues, this discrete approach could face challenges when applied to continuous transformations, such as scaling and shear.

5. Issues with Filter Generation from Rotated Inputs: Generating convolutional filters from rotated input images could present a problem. Since the translation-rotation group does not commute (i.e., rotation followed by translation differs from translation followed by rotation), G-CNNs typically rotate the convolutional filters rather than the input images. While the proposed method achieves translation-rotation equivariance, the approach of generating filters based on rotated inputs could introduce inconsistencies, as translation of the input image would alter the generated filters. Fortunately, the use of a regular CNN as the NEP helps maintain some robustness to translation. However, standard CNNs face challenges in achieving ideal translation equivariance [1]. While this paper focuses on rotation equivariance, neglecting translation equivariance could impact the proposed method more significantly than in other G-CNNs.

[1] Aharon Azulay and Yair Weiss. Why do deep convolutional networks generalize so poorly to small image transformations? Journal of Machine Learning Research, 20(184):1–25, 2019.

**Questions:**

1. What is the train/test split for the MNIST datasets used in this paper? Is it 12k/50k?

---

> ### Author Response · Authors · 2024-11-25
>
> Thank you for your constructive comments. Below we have made responses to your comments. If you have any further comments, please feel free to let us know and we are more than glad to discuss with you.
>
> > Question 1: What is the train/test split for the MNIST datasets used in this paper? Is it 12k/50k?
>
> For the original MNIST and the randomly rotated MNIST, we use the default split from torchvision MNIST dataset, which is 60k/10k for training and testing, respectively.
>
> >Weakness 1: Outdated and Inconsistent Baselines.
>
> We want to clarify that our main contribution is to propose the HE-CNN, achieving equivariance with a novel approach and alleviating the constraints presented in methods based on G-CNN. Thus, main baseline methods in comparison are G-CNN-based methods such as partial equivariance [1]. Since most equivariance models such as steerable CNN are based on G-CNN, we believe that our proposed method has lots of potential and extensions.
>    Steerable CNN, as a more generalized version of G-CNN, is very powerful, and our method shows comparable performance with Steerable CNNs. In our future work, we will apply our method to steerable CNNs.
>
> > Weakness 2 and 3: Missing Parameter Counts, and Questionable Results in Table 3.
>
> In the following table, we provide the comparison among different models in terms of the number of parameters. We agree with you that the performance with LoRA is a more fair comparison. We never claim to be the new state-of-the-art in our paper, and we have further clarify that in the paper to avoid misunderstanding.
>
> | Model        | Parameter Count |
> | ------------ | :---------------:|
> | Base CNN     | 28.6k           |
> | G-CNN        | 98.0k           |
> | Sim2-CNN     | 864k            |
> | HE-CNN       | 9.68m           |
> | HE-CNN(LoRA) | 51.9k           |
>
> Additionally, for the 13-layer-CNN on CIFAR10, we provide a comparison of the number of parameters and training time.
>
> | Model Trained with CIFAR10 | Test Accuracy(%) | Parameters | Training Time per iteration |
> | --- | ---: | ---: | ---: |
> | 13-Layer CNN (Base) | 91.21 | 3,119,754 | 18.67s/it |
> | Z4-CNN | 89.73 | 12,456,586 | 87.92s/it |
> | Z4-HE-CNN | 91.95 | 380,271,050 | 151.77s/it |
> | Z4-HE-CNN (LoRA+Head) | 90.12 | 23,129,268 | 139.82s/it |
> | Z4-CNN trained with Dynamic Hypernet w/ LoRA+Head | 89.84 | 32,627,215 | 147.21s/it |
>
> >Weakness 4: Limited Generalizability to Other Transformations.
> >
> In section 4.5, we expand to 90/n-degree rotations and reflection, which is the same as G-CNN. However, G-CNN has managed to extend to Steerable CNN which is more versatile in terms of chosen groups. Although we are still limited to certain groups, we believe that our HE-CNN has the potential of extending in our future works through the study of group theory and parameter space symmetry.
>
> > Weakness 5: Issues with Filter Generation from Rotated Inputs, and concerns on translation robustness.
>
> We would like to point out that our HE-CNN imposes no restrictions on the generated filters for images with the same labels. For example, two different images of "7" may produce different sets of filters. If two images are exactly rotated versions of each other, their generated filters will also be related through the same rotation, guaranteed through the equi-combiner.
>
> To address your concern on small translations, we provide an additional experiment. For our model trained with Rot-MNIST in section 5.2, we add random translation augmentation to the Rot-MNIST testing set and then test each trained model on it. (No translation augmentation is added during training.) We observe that your assumption is correct: $Z4$-CNN is more robust to translation, with performance dropping from 97.72% to 42.89%. Our method, dropping from 97.91% to 29.40%, exhibited a similar performance decline compared to the regular CNN structure. We thus conclude that our method is as robust as the regular CNN. As you mentioned, our motivation did not include translation invariance, and in future works, we aim to look deeper into this question and study translation equivariance in greater details.
>
>
> | Model |  Accuracy (%) with Translation: |
> | -------- |  -------- |
> | Regular CNN     |   26.61%  |
> | $Z4$-CNN     |     42.89%  |
> | $Z4$-HE-CNN(LoRA)     |        29.40% |
>
> >[1] Romero, D. W., & Lohit, S. (2021). Learning partial equivariances from data. arXiv preprint arXiv:2110.10211.
> >
> >[2] Aharon Azulay and Yair Weiss. Why do deep convolutional networks generalize so poorly to small image transformations? Journal of Machine Learning Research, 20(184):1–25, 2019.

---

> ### Comment · Reviewer_TBde · 2024-11-28
>
> > To address your concern on small translations, we provide an additional experiment. For our model trained with Rot-MNIST in section 5.2, we add random translation augmentation to the Rot-MNIST testing set and then test each trained model on it. (No translation augmentation is added during training.) We observe that your assumption is correct: CNN is more robust to translation, with performance dropping from 97.72% to 42.89%. Our method, dropping from 97.91% to 29.40%, exhibited a similar performance decline compared to the regular CNN structure. We thus conclude that our method is as robust as the regular CNN. As you mentioned, our motivation did not include translation invariance, and in future works, we aim to look deeper into this question and study translation equivariance in greater details.
>
> I appreciate the extra experiment that confirms my thought.
>
> The proposed method achieves rotation equivariance because the group has the semidirect product structure: $(R^2, +) \rtimes G$, where $G$ is Z4 or Dn in the context of this paper. The method uses the NEP generator and Equi-Combinator to take care of the transformations in group $G$, and the main network to take care of the translation in $R^2$. The decrease of accuracy for both G-CNN and HE-CNN is caused by the imperfect translation equivariance. Note that G-CNN is affected by this issue only once whereas twice for HE-CNN. One in the NEP generator and one in the main network. Thus, the same issue will arise even if HE-CNN is upgraded to incorporate steerable CNNs.
>
> The paper argues that the HE-CNN can reduce constraints imposed on filters so that HE-CNN is more robust to little rotational variation than G-CNN and more versatile . I think, what is happening is that the constraints imposed on filters w.r.t. the group $G$ are transferred to the constraints imposed on filters w.r.t. $(R^2, +)$.
>
> This method would be better off with CNNs having improved translation equivariance in the NEP generator. However, this means that the authors need to redo all experiments. Alternatively, the paper should show a trade-off between being versatile and being vulnerable to translation errors (with more parameters than baselines). Unfortunately, I don't see current experiments can show such a trade-off.

---

> > ### Author Response · Authors · 2024-12-02
> >
> > We thank the reviewer m86f for your constructive replies. Below, we provide responses to your feedback. If you still have any questions or need further clarification, please feel free to let us know, and we would be happy to clarify.
> >
> > > The reviewer is still concerned about our model's performance with small translations. As shown in the experiments from our previous response, our model demonstrated similar robustness against random translations, while G-CNN exhibited better robustness in this area.
> >
> > We appreciate the reviewer for sharing their concerns regarding the robustness of our translations. We agree that our model currently demonstrates less robustness than G-CNN. However, due to the versatility of hypernetworks, we believe this problem can be partially addressed by improving the parameter generation process. By studying which types of dynamic filters are beneficial for translations, we think better performance can be achieved by incorporating specific loss functions. In our future work, we look forward to investigating this area further, especially when we are not primarily focused on rotational equivariance.

---

> > > ### Comment · Reviewer_TBde · 2024-12-02
> > >
> > > As group convolution does not commute (a * b != b * a), there was a debate in [1] about the merits and drawbacks of rotating feature maps (inputs) versus rotating kernels. Both [1] and this paper adopt the former approach, while researchers following Cohen and Welling (2016) have chosen the latter. Notably, the weaknesses of the kernel rotation approach have been addressed and significantly improved upon in subsequent research.
> > >
> > > My concerns about rotating feature maps (or inputs) extend beyond robustness relative to translation. There are additional challenges, such as:
> > >
> > > * Handling non-square images: Rotating non-square images to achieve rotation equivariance introduces complications.
> > >
> > > * Scaling for scale equivariance: When sampling scale factors, extreme scaling (either large or small) can result in objects going out of image boundaries, causing information loss. These issues are not present in the kernel rotation approach.
> > >
> > > I have increased the score to 5 because the authors have honestly acknowledged the weaknesses of their approach. This transparency is appreciated and contributes to a more balanced evaluation of the work.
> > >
> > > [1] Dieleman, Sander, Jeffrey De Fauw, and Koray Kavukcuoglu. "Exploiting cyclic symmetry in convolutional neural networks." International conference on machine learning. PMLR, 2016.

---

> > > > ### Author Response · Authors · 2024-12-04
> > > >
> > > > Thank you for increasing the score from 3 to 5; we sincerely appreciate your constructive feedbacks. We are committed to addressing potential issues we discussed and will actively pursue solutions to enhance the quality of our work. Thank you once again for your replies.

---

### Author Response · Authors · 2024-12-04
**General Response**

Dear Area Chair,

As the authors of this paper, we appreciate the opportunity to present our current position on our paper.

We thank all reviewers for taking the time to review our work and giving us constructive and valuable comments to improve the paper.
1. All reviewers acknowledged the novelty of the proposed HE-CNN framework (Reviewers `TBde`, `m86f`, `kvp3`, and `JNi2`) as an innovative approach to achieving equivariance in CNNs without the use of any $G$-CNN structures, while outperforming $G$-CNN across all tested datasets.
2. As discussed in our paper, previous $G$-CNNs impose constraints on the filters (Further discussed with reviewer `JNi2`). We conducted an ablation study (Suggested by reviewer `JNi2` and `m86f`) that empirically demonstrates how our approach relaxes those constraints, improving performance on datasets with minimal rotational variations (as noted by Reviewers `TBde` and `JNi2`). When compared to $G$-CNN-based state-of-the-art methods, we showed comparable results (Reviewers `m86f`, `kvp3`, and `JNi2`).
3. We discussed the limitations of our method with the reviewers, including scalability issues (Reviewers `TBde`, `m86f`, `kvp3`, and `JNi2`), robustness to small translations (Reviewer `TBde`), and a relaxed definition of equivariance (Reviewer `m86f`).

During the rebuttal period, we responded to all the comments of all the reviewers.

Thank you once again for your kind consideration of our work.

Best regards,
Authors

---

### Meta-Review · Area_Chair_Lbpk · 2024-12-18

**Metareview:**

This paper introduces a hypernetwork-based approach to achieving rotational equivariance in CNNs, referred to as HE-CNN. The main idea is to equip a standard CNN with filters generated by a separate hypernetwork, ensuring that these filters transform in a manner consistent with a chosen rotation group. Instead of imposing fixed group-theoretic constraints on the filters, this approach leverages a neural parameter generator and an “equi-combiner” to produce filters that exhibit rotational equivariance by construction.

While the proposed idea of achieving equivariance by generating filters from a hypernetwork rather than restricting the filter space is interesting and reasonable, there were several critical concerns raised by the reviewers. It includes (1) scalability issue, since the method requires passing each input through the NEP generator for each group element, which scales poorly for larger groups, larger images, or large datasets, (2) extensive parameter counts, since the method can have orders of magnitude more parameters than baseline CNNs and G-CNNs, (3) missing comparisons to stronger baselines, (4) limited practicality, since the method requires all examples in the batch share the same orientation. While some minor concerns have been addressed by the authors’ response, these critical concerns remain unresolved.

After reading the paper, reviews, and discussions, the AC agrees with the reviewers that the paper has potential but is insufficient to be published in its current form. The AC recommends rejection this time.

**Additional Comments On Reviewer Discussion:**

The four major weaknesses in the meta-review were not successfully addressed by the rebuttal.

---

### Decision · Program_Chairs · 2025-01-22

Reject